# USP2-Related Cellular Signaling and Consequent Pathophysiological Outcomes

**DOI:** 10.3390/ijms22031209

**Published:** 2021-01-26

**Authors:** Hiroshi Kitamura, Mayuko Hashimoto

**Affiliations:** Laboratory of Veterinary Physiology, School of Veterinary Medicine, Rakuno Gakuen University, Ebetsu, Hokkaido 069-8501, Japan; s21841005@stu.rakuno.ac.jp

**Keywords:** ubiquitin-specific protease, tumorigenesis, cell cycle, inflammation, circadian clock, metabolic disorder, brain, fertility, deubiquitination, ion balance

## Abstract

Ubiquitin specific protease (USP) 2 is a multifunctional deubiquitinating enzyme. USP2 modulates cell cycle progression, and therefore carcinogenesis, via the deubiquitination of cyclins and Aurora-A. Other tumorigenic molecules, including epidermal growth factor and fatty acid synthase, are also targets for USP2. USP2 additionally prevents p53 signaling. On the other hand, USP2 functions as a key component of the CLOCK/BMAL1 complex and participates in rhythmic gene expression in the suprachiasmatic nucleus and liver. USP2 variants influence energy metabolism by controlling hepatic gluconeogenesis, hepatic cholesterol uptake, adipose tissue inflammation, and subsequent systemic insulin sensitivity. USP2 also has the potential to promote surface expression of ion channels in renal and intestinal epithelial cells. In addition to modifying the production of cytokines in immune cells, USP2 also modulates the signaling molecules that are involved in cytokine signaling in the target cells. *Usp2* knockout mice exhibit changes in locomotion and male fertility, which suggest roles for USP2 in the central nervous system and male genital tract, respectively. In this review, we summarize the cellular events with USP2 contributions and list the signaling molecules that are upstream or downstream of USP2. Additionally, we describe phenotypic differences found in the in vitro and in vivo experimental models.

## 1. Introduction

Protein ubiquitination and deubiquitination are reversible processes that control the fate of target proteins and protein–protein interactions. Deubiquitinating enzymes (DUBs), which are encoded by ~100 loci in the human genome, are either cysteine- or metalloproteases [1]. DUBs can be classified into five subfamilies: ubiquitin-specific proteases (USPs), ubiquitin C-terminal hydrolases, ovarian tumor proteases, Machado-Joseph disease proteases, and JAB1/MPN/Mov34 metalloproteases [2]. Of these, USPs constitute the largest DUB family, comprising ~60 members in vertebrates [3]. USP2 is the second member of the USP family and was originally identified in 1997 as UBP41 in chicken muscle [4]. In 2000, Lin et al. demonstrated that orthologues of UBP41 were exclusively expressed in mouse testis [5]. Four distinct splicing variants for *Homo sapiens* have been deposited in the UniProt protein database (accession number: O75604). Reglinski et al. defined the four isoforms as isoforms 1, 2, 3, and 4, which comprise 605, 353, 363, and 397 amino acids, respectively [6]. The N-terminal sequences of each isoform are as follows: isoform 1, MSQLSSTL……SSPGRDGM; isoform 2, MLNKAK; isoform 3, MLVPGSTRPSSKKR; and isoform 4, MRTSYTVT…..GLLLNKA [6]. Based on the amino acid sequence, isoform 1 (orthologue of human isoform 1), isoforms 2 and 3 (orthologues of human isoform 4), and isoform 3 (not deposited in human data) are present in the mouse data (O88623); however, one of the initial reports showed the existence of a mouse orthologue of human isoform 2 [7]. Human and mouse isoform 1 have been commonly referred to as USP2-69 or USP2A, while the nomenclature of the other isoforms, especially isoforms 2–4, has not been unified in the literature or databases [6,8]. In this review, we refer to the variants according to the nomenclature used by UniProt (O75604) to avoid confusion: isoforms 1, 2, 3, and 4 in human and their mouse orthologues are described as USP2-1, -2, -3, and -4. The C-terminal cysteine protease domains in these USP2 variants are identical, whereas their N-terminal extensions exhibit distinct structures. The different N-terminal structures have been postulated to interact with diverse association partners, resulting in distinct subcellular localizations and cellular events [5,9,10,11,12]. As mentioned above, although USP2 was once thought to be exclusively expressed in testis [12], it has also found to be abundant in a wide variety of cells and tissues, including the liver, heart, brain, skeletal muscle, kidney, and macrophages [9,12,13]. Some variant-specific roles have been reported, with certain USP2 variants being crucial for various physiological and pathological phenomena including tumorigenesis, circadian rhythm regulation, and inflammation. By comparing USP2 with other DUBs, some review articles have summarized the pathological roles of USP2 in specific functional areas, such as cancer promotion [14,15], muscle atrophy modification [16], and sodium channel regulation [17]. Additionally, another review focused on the expressional control of alternative splicing variants of USP2 [8]. Despite these examples in the literature, there have been no review articles summarizing USP2-associated signaling and its outcomes from a more comprehensive viewpoint. In this review, therefore, we examine the pathophysiological events elicited by USP2 and summarize the cellular signaling that underlies the events in a comprehensive way.

## 2. Tumorigenesis

The most documented topic associated with USP2 is tumorigenesis. To date, various types of malignant tumors, including prostate cancer, hepatoma, bladder carcinoma, and glioma, have been reported to express high levels of USP2 [18,19,20,21,22]. Additionally, triple-negative breast cancer, which is the most aggressive type of breast cancer, exhibits high levels of USP2 expression in conjunction with enhanced cell migration and invasion [23]. Accordingly, forced expression of *USP2-1* in cultured bladder cancer cells was shown to cause proliferation, invasion, migration, and enhanced resistance to chemotherapy [24]. A recent paper demonstrated that USP2-1 was upregulated in the stem cells of triple-negative breast cancer, where it supported their maintenance by activating self-renewal [25].

Several molecular targets of USP2 have been found in cancer cells. Fatty acid synthase (FASN), which synthesizes palmitate from acetyl-CoA and malonyl-CoA, is frequently overexpressed in tumor cells, where it inhibits apoptosis [18,19,26]. USP2-1 potentiates the stability of FASN by inhibiting proteasome-dependent degradation in prostate cancer [18], hepatoma [19], mantle cell lymphoma [26], and glioma [20]. Additionally, overexpression of myristoylated Akt increased USP2-1 expression in hepatoma, which was accompanied by high levels of expression of lipogenic proteins including FASN, acetyl-CoA carboxylase, and adenosine triphosphate citrate lyase [19]. These results imply that the acceleration of lipogenesis may be the cause of USP2-1-elicited tumorigenesis.

A canonical tumor suppressor, p53, is another target of USP2. Under conditions with cellular stress and/or DNA damage, p53 accumulates in cells and evokes various anti-tumor events such as DNA repair, induction of apoptosis, or cell cycle arrest [27,28]. Loss-of-function of p53 is now believed to be one of the common events underlying tumorigenesis [29]. The cellular content of p53 is controlled at both the transcriptional and post-transcriptional levels. Murine double minute (MDM) 2, an E3-ubiquitin ligase, promotes the elimination of p53 by a proteasome-dependent pathway [30]. USP2-1 stabilizes MDM2 via the de-ubiquitination of its poly-ubiquitin chain, which decreases the levels of intracellular p53 in prostate cancer and cutaneous T-lymphomas [15,31,32]. MDMX is another repressor of p53, and attenuates the expression of p53 downstream genes by binding the active transcription site of p53 [33]. A previous paper demonstrated that USP2-1 stabilized MDMX and increased cell survival when cultured tumor cells were treated with cisplatin [15,34]. USP2 also controls MDM4, which shares structural similarity with MDM2 [30]. Additionally, USP2 directly stabilizes MDM4 in the cytoplasm of gliomas [35]. In turn, accumulated MDM4 conveys p53 to mitochondria, which is followed by the promotion of cytochrome *c*-inducible apoptosis [35]. On the other hand, USP2 has also been shown to increase p53 in a hepatoma cell line (HepG2) and a breast cancer cell line (MCF7) after leptin stimulation [36].

The augmentation of cell cycle progression is a common feature of cancer cells, and the roles of USP2 in cell cycle regulation have been relatively well-studied. The aberrant overexpression of cyclin D1 is frequently observed in various types of cancerous cells [37,38]. By screening using cyclin D1 as a substrate, Shan et al. identified USP2 as a specific cyclin D1 deubiquitinating enzyme among 76 DUBs [39]. USP2 directly interacted with cyclin D1 and decreased polyubiquitination-dependent degradation [39]. Moreover, USP2-1 is a target of a lithocholic acid hydroxyamide, which also destabilizes cyclin D1 [40]. In both hepatoma and breast cancer cell lines, leptin causes cell cycle progression and adiponectin causes cell cycle arrest [41]. Since leptin and adiponectin have opposite effects on USP2 expression in these cells, USP2 likely contributes to cell cycle regulation via adipokines [41]. Accordingly, the overexpression or knockdown of *USP2* has been shown to modulate cyclin D1 expression, which is increased and decreased by leptin and adiponectin, respectively [41]. Hence, these adipokines appear to modify USP2 expression in a manner that leads to changes in cyclin D1 levels and subsequent cell cycle progression. Since the intracellular content of cyclin D1 is a determinant for tumorigenesis in certain types of tumors, cyclin D1 has been used as an index in the exploration of USP2 inhibitors as anti-tumor drugs [40,42,43]. Similar to cyclin D1, USP2-1 likewise stabilizes cyclin A1, which also participates in the proliferation of bladder cancer cells [24]. 

Aurora-A, a serine/threonine kinase, is known to be vital for centrosome duplication and maturation, but excessive expression of Aurora-A causes instability of genomic DNA, leading to oncogenesis [44,45]. Shi et al. demonstrated that USP2-1 directly deubiquitinates Aurora-A, and thus stimulates mitotic progression of MIA PaCa-2 pancreatic carcinoma cells [46]. In agreement, the attenuation of proliferation by the administration of *USP2-1* small interfering RNA (siRNA) was found to be overwhelmed by ectopic expression of Aurora-A in cells [46]. Thus, USP2-1 seems to stimulate mitosis in pancreatic carcinoma by stabilizing Aurora-A.

The epithelial–mesenchymal transition (EMT) is a triggering event for tumor metastasis [47]. EMT-associated genes are induced by transforming growth factor (TGF) β–signaling [48]. After binding of TGF-β to heterodimeric TGF-β receptors, receptor-regulated SMADs (R-SMADs) are recruited to the intracellular domain of the receptors [49]. Subsequently, R-SMADs dissociate from the ligand-receptor complex after serine is phosphorylated by the receptor, and then associate with SMAD4 in the cytoplasm [50]. The complex consisting of R-SMADs and SMAD4 then enters the nucleus, and initiates the transcription of EMT-associated genes [51]. USP2 facilitates the binding of R-SMADs to TGF-β receptors by removing the K33-linked poly-ubiquitin chains on TGF-β receptors [52]. Accordingly, a selective USP2 inhibitor, ML364, has been shown to effectively suppress tumor metastasis in mice [52].

Wnt/β-catenin signaling also plays a pivotal role in the EMT [53,54,55,56]. Although β-catenin functions as a component of adherens junctions, nuclear β-catenin augments the transcription of EMT-related genes by interacting with several transcription factors, such as DNA-bound T cell factor/lymphoid enhancer factor [56]. In normal epithelial cells, β-catenin is continuously digested by a proteasome-dependent mechanism because the glycogen synthase kinase (GSK)3β /Axin/adenomatous polyposis coli (APC) complex promotes the polyubiquitination of β-catenin [57]. After the binding of Wnt to the G protein–coupled receptor Frizzled, Dishevelled proteins inactivate the GSK3β/Axin/APC complex, which results in the accumulation of β-catenin in the cytoplasm [57]. Consequently, β-catenin translocates into the nucleus and initiates the transcription of EMT-related genes. In a previous paper, the screening of β-catenin deubiquitinase using expression constructs encoding 68 human deubiquitinases and showed that USP2-1 directly interacts with β-catenin, which results in an increase in the level of β-catenin protein via a direct interaction [58]. Coincidently, genetic and pharmacological inhibition of USP2 also decrease β-catenin and attenuate β-catenin-dependent gene expression [58]. 

Erythroblastic oncogene B2 (ERBB2)/human epidermal growth factor receptor 2 (HER2) is a receptor tyrosine kinase and a member of the epidermal growth factor (EGF) receptor family. ERBB2 overexpression is highly associated with poor prognosis in breast cancer, and the Food and Drug Administration has designated ERBB2 as an efficient therapeutic target [59]. Inhibitors of heat shock protein (HSP) 90 foster the polyubiquitination of ERBB2, which leads to the cleavage of full-length ERBB2 into a signaling-impaired fragment in the early endosome [60]. Thus, several HSP90 inhibitors, such as geldanamycin, ganetespib, neratinib, tanespimycin, and alvespimycin, have been given to patients with breast cancer [60,61,62]. USP2 maintains ERBB2 levels by counteracting endocytic degradation [62]. Thus, combinatorial treatment using HSP90 inhibitors and USP2 inhibitors is suitable for ERBB2-elicited tumorigenesis. Given that ERBB2 is not only involved in tumorigenesis, but also in physiological events such as neural repair [63], USP2 may also modulate ERBB2-associated physiological events. In addition to ERBB2, the EGF receptor (EGFR) is also proposed to be controlled by USP2. Surface expression of EGFR is strongly regulated by internalization [64], and the impairment of this endocytic mechanism causes constitutive activation of EGF signaling and carcinogenesis [64,65]. In lung cancer cells, USP2 is distributed to the early endosome, and removes the polyubiquitin chain from internalized EGFR in early endosomes [66]. Thus, USP2 has the potential to increase EGFR content on the cell surface.

Acid ceramidase (ACDase) is an enzyme that synthesizes sphingosine from ceramide under acidic conditions [67]. Sphingosine is utilized to produce sphingosine-1 phosphate, which promotes cell survival, proliferation, migration, and invasion in cancer [68]. ACDase is markedly accumulated in several malignant cancers, including prostate cancer [69]. Murate and his colleagues reported that the expression of *ASAH1* mRNA, which encodes ACDase, did not correlate with ACDase protein levels in an androgen-sensitive prostate cancer cell line (LNCaP); however, administration of either an androgen receptor antagonist or charcoal-stripped serum decreased ACDase protein levels via a proteasome-dependent mechanism [70]. Furthermore, the overexpression or knockdown of *USP2* caused a respective increase or decrease in ACDase in the LNCaP cells [70]. Given that USP2 is upregulated by androgen [18], USP2 may therefore promote the onset of androgen-sensitive prostate cancer via the accumulation of ACDase.

The maintenance of mitochondrial integrity by USP2-1 is another potential mechanism that explains USP2-1–dependent tumor cell survival. USP2-1 downregulates miR-34b/c, which leads to the increase in c-Myc in prostate cancer cells and prostate epithelial cells [71,72]. In turn, c-Myc induces the expression of γ-glutamyl-cysteine synthase, which is a rate-limiting enzyme for glutathione synthesis. Glutathione reduces reactive oxygen species (ROS) and stabilizes mitochondrial membrane potential [71]. Therefore, USP2-1 confers resistance against pro-oxidant anti-tumor drugs, such as cisplatin and doxorubicin, via the miR-34b/c–driven c-Myc pathway [71]

Although the majority of the literature indicates a positive correlation between USP2-1 expression and tumorigenesis, several reports have also shown that the *USP2* mRNA level is decreased in some types of tumors. For example, a comprehensive gene expression analysis of 18 human cancer types from the Cancer Genome Atlas categorized *USP2* under “consistently downregulated genes” in adrenocortical carcinoma, kidney renal clear cell carcinoma, and thymoma [73]. Accordingly, the overexpression of USP2 in clear renal carcinoma cells mitigated proliferation, cell migration, and invasion, but the underlying cellular signaling is still elusive [74].

## 3. Apoptosis and Autophagy

In an early report, Gewies et al. indicated that the overexpression of UBP41 (which, based on the amino acid sequence, may be USP2-2) stimulated apoptosis and caspase-3 activation in HeLa cells [7]. In sharp contrast, Priolo et al. demonstrated that *USP2-1* siRNA increased the apoptosis rate in the prostate cancer cell lines LNCaP (androgen dependent, wild type p53) and DU145 (androgen independent, mutant p53), but not PC-3 (androgen independent, p53-null), suggesting an anti-apoptotic role of USP2-1 [75]. That report further showed that *USP2-1* siRNA promoted the apoptotic response of nine non-prostate human tumor cell lines, which were derived from colon cancer, breast cancer, and sarcoma [75]. Additionally, USP2-1 conferred chemoresistance to cisplatin and paclitaxel in prostate epithelia-derived AR-iPrEC cells [75]. In particular, *USP2-1* knockdown increased the levels of p53, and its target, p21, but significantly reduced the levels of FASN and MDM2 [75]. USP2-1 also stabilized MDMX and MDM4, both of which are known to be negative regulators of p53 [34,35]. Given the proapoptotic properties of p53, it can be concluded that USP2-1 abates apoptosis by stabilizing MDM2, MDMX, MDM4, and FASN. 

In addition to cancer cells, USP2 has been also postulated to modulate tumor necrosis factor (TNF) α–induced apoptosis in hepatocytes. Haimerl et al. showed that pre-treatment with a low dose of TNF-α confers TNF resistance to hepatocytes, both in vivo and in vitro [13]. In the same paper, the authors also established USP2-2 as the dominant USP2 isoform in the liver, and demonstrated that USP2-2 was dramatically downregulated in the livers of mice after treatment with a low dose of TNF-α [13]. This finding suggests that a reduction in USP2 levels is necessary for the acquisition of TNF resistance in hepatocytes. The observation that the overexpression of USP2-2 ameliorated the beneficial effects of the TNF-α pretreatment, while USP2-2 also promoted hepatocyte death during post-treatment with actinomycin D (ActD) and TNF-α, further supports this idea [13]. Conversely, *USP2* siRNA inhibited ActD/TNF-α−elicited apoptosis via decreased formation of active caspase-3 [13]. *USP2* knockdown also evoked an increase in anti-apoptotic protein cellular FLICE-like inhibitory protein (cFLIP) and a concomitant decrease in the E3-ubiquitin protein ligase Itchy homolog (ITCH) [13]. On the other hand, overexpression of USP2-2 decreased the levels of ubiquitinated ITCH protein, which consequently led to the accumulation of ITCH in isolated hepatocytes, especially after treatment with MG132 [13]. Because cFLIP is believed to be one of the critical factors for TNF-α–induced cell survival, USP2-2 may therefore inhibit TNF-resistance via the promotion of ITCH-mediated cFLIP degradation [13]. 

After TNF-α binds to TNF receptor 1 (TNFR1), the TNF receptor 1 complex II leaves TNFR1 and subsequently activates the caspase cascade [76]. Mahul-Meiller et al. demonstrated the modulatory role of USP2 for signaling molecules in the TNFR1-triggered apoptosis cascade [77,78]. In MCF7 and HEK293-T cells, TNF-α stimulation led to USP2-1 associating with the TNFR1 complex I, involving receptor-interacting serine/threonine-protein kinase 1 (RIP1) and TNF receptor-associated factor (TRAF) 2 [77]. Overexpression of *USP2-1* removed the K63-linked polyubiquitin chains on RIP1 and TRAF2, and promoted the K48-linked polyubiquitination of RIP1, but not TRAF2 [77]. On the contrary, *USP2* knockdown augmented the K63-linked polyubiquitination of both RIP1 and TRAF2 [77]. Moreover, USP2 deficiency also caused a prolonged formation of the TNFR1 complex I with concomitant decrements of TNFR1 complex II and cleaved caspase-8 [77]. Therefore, USP2 may facilitate the formation of TNFR1 complex II from complex I by removing the polyubiquitin chains on RIP1 and TRAF2. On the other hand, *USP2* knockdown sustained the IκBα degradation associated with prolonged activation of p38 and c-Jun N-terminal kinase (JNK) [77]. Thus, USP2 is likely to be a factor responsible for TNF signaling leading to apoptosis, but not for NF-κB-dependent cell survival. The same authors also demonstrated that the ratio of USP2-1 and TRAF2 determined the apoptosis sensitivity in response to TNF-α, whereby high USP2-1/TRAF2 conditions accelerated TNF-induced cell death, and low USP2-1/TRAF2 conditions inhibited death [78]. In addition to USP2-1, USP2-2 also caused caspase-8 activation, cleavage of poly ADP-ribose polymerase (PARP), and promotion of cell death [78]. As with USP2-1, USP2-2 overexpression substantially increased RIP1 levels [78]. Conversely, *RIPK1* (gene symbol for RIP1) knockdown abated USP2-2-induced cell death, indicating that RIP1 is downstream of USP2-2 [78]. Indeed, USP2-2 was found to associate directly with RIP1 and removed its K48- and K63-linked polyubiquitin chains [78]. USP2-2 also digested the K63-polyubiquitination of TRAF2, but not its K48-linked chain, suggesting TRAF2 as another putative target of USP2-2 [78]. Interestingly, not only overexpression, but also ablation of *USP2-2* mRNA, precipitated cell death [78]. Further studies are required to verify how the abundance of USP2-2 regulates cell death.

Possible roles of USP2 in autophagy have also been recently reported. Sorafenib has been used to treat advanced stage hepatocellular carcinoma (HCC) [79]; however, sorafenib resistance is frequently acquired and is one of the most severe challenges in treating HCC [80]. A sorafenib-resistant Huh7-derived clone exhibited no obvious activation of caspase-3 and PARP, concomitant with a decreased rate of apoptosis [81]. Endoplasmic reticulum stress (ERS)-related proteins, such as protein kinase RNA-like ER kinase, ER oxidoreductin-1-like α, and binding immunoglobulin protein were abundantly expressed in these clones and were all downregulated in response to treatment with ERS inhibitor 4-phenylbutyric acid (4-PBA). Treatment with 4-PBA also restored sorafenib sensitivity by attenuating cell viability and activating caspases, and inhibited the increased progression of autophagosome formation in these sorafenib-resistant clones [81]. Conversely, blocking autophagosome formation clearly restored sorafenib sensitivity [81]. The sorafenib-resistant clones showed high expression of cFLIP, which potentiates ERS induction and suppression of apoptosis [81]. Compared to its parental cells, the sorafenib-resistant clones express lower levels of USP2, which presumably caused the aforementioned increase in cFLIP levels and decrease in ITCH levels [13,81]. Indeed, the restoration of USP2 expression in the sorafenib-resistant clones prevented the elevation of cFLIP levels and decreased ITCH levels [81]. Collectively, USP2 potentially modulates autophagy and ERS by altering the abundance of cFLIP.

An aspect of subcellular USP2 localization has also been explored as a mechanistic explanation for the inhibition of USP2-elicited cell death. USP2 has a potential peroxisomal targeting signal 1 (PTS1) signal on its C-terminus that comprises a consensus sequence of three amino acids: (S/A/C)(K/H/R)(L/M) [6,82]. A previous paper demonstrated that peroxisomal import regulates the proapoptotic activity of USP2 [6]. Overexpression of *USP2-3* elicited the most dramatic increase in caspase activity (~2.2-fold), whereas the other three isoforms caused 1.6 to 1.8-fold increases in HEK293 cells [6]. The proapoptotic activity of USP2-3 was not inhibited by the deletion of the PTS1 signal [6]. In stark contrast, substitution of the PTS1 motif of USP2 with the optimized PTS1 signal strongly repressed caspase activity and led to a prominent localization of USP2-3 in the peroxisome [6]. Since peroxisome import is much faster and requires less energy than proteasome-dependent ablation, the authors hypothesized that the peroxisome works as a “metabolic valve” for the supply of proapoptotic USP2, which immediately determines cell fate [6].

## 4. Circadian Clock

The circadian clock is generated by several molecular clock proteins, including brain and muscle Arnt-like protein 1 (BMAL1), clock circadian regulator (CLOCK), period (PER), cryptochromes (CRY), REV-ERB, and retinoic acid receptor-related orphan receptor (ROR) [83,84]. A heterodimer that comprises BMAL1 and CLOCK activates the transcription of the genes encoding the PER and CRY proteins. Subsequently, PER and CRY repress the transcription of BMAL1/CLOCK [84]. In addition, ROR and REV-ERB positively and negatively modulate the transcription of the *BMAL1* gene [84]. Therefore, the abundance of these molecular clock proteins is a result of transcription-dependent synthesis strictly counterbalanced with proteasome-dependent degradation [84]. A previous transcriptomics analysis revealed that *USP2* was among the ten genes that were expressed in a circadian manner in all organs [85]. Accumulating evidence further indicated that USP2 modulated the circadian molecular clock in mice, but no severe circadian deficits were found in *Usp2* knockout (*Usp2*KO) mice under normal 12-hour light/dark cycle conditions [86] (Figure 1).

Mammals can adjust circadian activities in response to irradiative light strength. The role of USP2 in circadian sensitivity to light has been also evaluated using *Usp2*KO mouse models. When *Usp2*KO mice were placed in constant light with low (but not high) irradiance after 12-hour light/dark conditions, the mice displayed a remarkable delay in the onset of the active phase [87]. Since mice increase or decrease their rhythmic activity in response to constant light with lower or higher irradiance, the delayed onset of the active phase in *Usp2*KO mice suggests that USP2 deficiency elevates sensitivity to low irradiative light. To support this, exposure to low irradiative light also caused a delay in the onset of the active phase after mice were housed in constant darkness [87]. In agreement, another paper also reported that *Usp2*KO mice required a longer time to adjust their circadian rhythm in response to changes in the light-dark cycle, and that they demonstrated aberrant responses to short light pulses administered across the daily cycle [86]. Additionally, the levels of PER1, an isoform PER, suprachiasmatic nucleus (SCN) of the hypothalamus, which is the dominant endogenous pacemaker, were markedly decreased in the *Usp2*KO mice [87]. All these findings suggest crucial roles for USP2 in the regulation of circadian rhythms. The sumoylation of BMAL1 in the promyelocytic leukemia nuclear body is a prerequisite for its ubiquitination in fibroblasts [88]. Introduction of USP2 to fibroblasts promoted the digestion of the ubiquitin chain on BMAL1 and increased the number of nuclear foci with sumoylated BMAL1 [88]. Sumoylated and ubiquitinated BMAL1 is most abundantly accumulated in the nucleus when the promoter of *Per2*, gene for another isoform of PER, is active [88]. Hence, USP2 seems to augment BMAL1/CLOCK-elicited gene expression. In agreement, modulatory roles of USP2 on BMAL1 have also been observed in the SCN [87]. The expression of the USP2-1 and USP2-4 proteins in the SCN are rhythmic; both variants are abundant from night to early morning, but the accumulation of USP2-4 is more marked [87]. Furthermore, immunoprecipitation analysis indicated that USP2-4 forms a complex with BMAL1 in the SCN and thereby stabilizes BMAL1, not but CLOCK or PER1 [87]. Indeed, *Usp2*KO mice exhibited aberrant induction of the CLOCK/BMAL1-induced genes, such as *Per1*, *Rev-erba*, and *Dbp* (gene for albumin site D-binding protein), in conjunction with a significant decrease in BMAL1 levels, in response to low-light irradiation for four hours from the time when the lights were turned off [87]. Therefore, BMAL1 is a downstream signaling molecule of USP2-4.

In addition to BMAL1, PER1 is another target of USP2. Using co-immunoprecipitation, Yang et al. demonstrated that USP2-1 or USP2-4 were associated with PER1, PER2, CRY1, CRY2 and BMAL1 in HEK293 cells [86]. PER1 was the only clock component that USP2-1 and USP2-4 directly interacted with in vitro, which implies that the USP2 isoforms indirectly bind other clock components [86]. Accordingly, the overexpression of *USP2-1* or *USP2-4* led to a decrease in the ubiquitinated form of PER1 [86]. Moreover, mouse embryo fibroblasts (MEFs) derived from *Usp2*KO mice showed constitutively higher levels of polyubiquitinated PER1 than control MEFs, under both synchronized and unsynchronized culture conditions [86]. Interestingly, neither cycloheximide nor MG132 affected the stability of PER1, indicating that USP2 reduced the polyubiquitination of PER1 without significantly modifying its stability [86]. Although PER1 was distributed in both the cytoplasm and nucleus in *Usp2*KO MEFs, PER1 was preferentially localized in the nucleus of wild-type MEFs [89]. Therefore, USP2 may control the subcellular localization of PER1. Additionally, *Usp2*KO MEFs showed an advanced nuclear translocation and shortened nuclear retention of PER1 [89]. As a result, USP2 ablation enhanced the amplitude of expression of the core clock genes, except *Per1*, which subsequently disrupted the expression of clock-controlled genes [89]. Likewise, an advanced peak phase of nuclear localization and shortened nuclear retention were also evident in hepatocytes isolated from *Usp2*KO mice [89].

USP2 has also been demonstrated to participate in the control of rhythmic calcium ion uptake from the small intestine [90]. *Usp2*KO mice exhibited increased calcium excretion in urine, with reduced calcium excretion when fasting or on a calcium-restricted diet [90]. Proteomics data from the intestine revealed that the lack of USP2 strongly increased the abundance of Na^+^/H^+^ exchange regulatory cofactor 4 (NHERF4) [90], which is a known regulator of a calcium ion channel [91]. Furthermore, a co-expression analysis indicated that USP2-4, not but USP2-1, directly interacted with NHERF4 and clathrin heavy chains [90]. USP2-4 was abundantly expressed in the small intestine and exhibited a circadian pattern of expression, to which NHERF4 showed an antiphasic pattern [90]. Additionally, *Cry1* and *Cry2* double-knockout mice exhibited lower amplitudes of rhythmic *Usp2* expression in the intestine [90]. Notably, the circadian changes in membrane NHERF4 in these double-knockout mice were smaller than those of control mice [90]. Together, these results suggest that USP2-4 modulates the circadian rhythm of calcium permeability in the intestine via posttranslational control of NHERF4 at the plasma membrane. 

## 5. Renal System

Epithelial sodium channel (ENaC) predominantly contributes to trans-epithelial sodium reabsorption in the renal connecting and collecting tubes. E3-ubiquitin ligase neural precursor cell–expressed developmentally down-regulated protein (Nedd) 4-2 polyubiquitinates ENaC and directs it to proteasome-dependent digestion [92]. Activity of Nedd4-2 is tightly regulated by phosphorylation. Phosphorylation of Nedd4-2 is catalyzed by serum and glucocorticoid-regulated kinase, which is induced by steroid hormones such as aldosterone, an endogenous mineralocorticoid [92]. USP2 has been shown to counteract the effects of Nedd4-2 to maintain ENaC expression levels in kidney epithelial cells [93]. Furthermore, Oberfeld et al. reported that USP2-4 directly bound the cytoplasmic N-terminal of αENaC, which was dependent on the status of an ubiquitination site of ENaC [94]. USP2-4 also associated with the HECT domain of Nedd4-2 in a manner independent of catalytic activity [94]. Of the three ENaC subunits, the surface expression of γENaC was increased by USP2 because USP2 digested the polyubiquitin-made “internalization signal” on γENaC [95]. In support of this, another report also demonstrated that USP2-4 increased the surface abundance of ENaC by interfering with endocytosis, but not digestion, at the lysosome [96]. Additionally, USP2-4 also contributed to the proteolytic cleavage of the extracellular loop of αENaC, leading to ENaC activation [95,97]. By preventing both internalization and extracellular loop digestion, the overexpression of *USP2-4* in a cellular model caused more than a 20-fold activation of ENaC [95]. 

In addition to surface ENaC expression, USP2-4 also determines the stability of mineralocorticoid receptors (MRs). MRs are receptors for aldosterone and induce expression of ENaC in epithelial cells leading to the reabsorption of sodium and water, and the consequent regulation of body fluid volume and blood pressure [98]. In addition to binding to MRs, aldosterone stimulates the phosphorylation of MRs in an extracellular signal regulate kinase (ERK)-dependent fashion [99]. Phosphorylated MRs are subsequently monoubiquitinated, which allows interaction with tumor suppressor gene (TSG) 101. Interaction with TSG101 elevates the stability of MRs [99]. USP2-4 disrupted the association between MRs and TSG101 by inhibiting the monoubiquitination of MRs [100]. Instead, MRs were polyubiquitinated and then digested by proteasomes, and their overall transcriptional activity was suppressed [100]. Given that USP2-4 is strikingly induced by aldosterone [17], the induction of USP2-4 may therefore function as a node of the negative feedback loop for MR signaling.

The significant regulatory roles of USP2 on ENaC and MR expression suggest that USP2-4 is a key modulator for sodium distribution in the body, and therefore for blood pressure. This idea was supported by a report on the population-based genome-wide association study of 7,552 subjects in Korea, which showed that a synonymous single nucleotide polymorphism of the *USP2* gene was statistically associated with blood pressure [101]. On the other hand, a study in mice reported that varying the amount of dietary sodium (< 0.01% to 3.2%) for two weeks did not alter *Usp2* expression in the cortical collecting duct, suggesting that USP2 is unlikely to be involved in sodium homeostasis in mice [102]. *Usp2*KO mice did not exhibit impaired diurnal rhythms of osmolality or excretion of creatinine, sodium, and potassium, in response to either high- or low-salt dietary challenges [102]. Similarly, plasma sodium, potassium, and aldosterone levels were indistinguishable between wild type and *Usp2*KO mice [102]. Furthermore, *Usp2*KO mice showed no obvious alterations to systolic or diastolic blood pressure in response to low-, normal-, or high-salt diets [102]. Therefore, USP2 seems to be dispensable for the regulation of sodium balance and blood pressure. Further studies are required to elucidate whether USP2 plays a primary role in these processes under pathological and/or physiologically stressed conditions.

## 6. Energy Metabolism and Metabolic Disorders

As previously mentioned, several reports have demonstrated that USP2 preserves FASN expression in several cell types, which implies that USP2 has a potential role in the regulation of lipid metabolism [18,19,20,26]. Indeed, accumulating evidence indicates that USP2 is a key regulating enzyme for energy metabolism in both physiological and pathological conditions (Figure 2).

Under fasting conditions, hepatic gluconeogenesis is necessary to maintain blood glucose levels. Glucagon and adrenal glucocorticoids increase the levels of blood glucose by activating hepatic gluconeogenic enzymes, including phosphoenolpyruvate carboxykinase (PEPCK) and glucose-6-phosphate (G-6-Pase) [103,104]. Because the circadian rhythm of the liver clock critically affects overall energy metabolism, the mechanisms that modulate the hepatic clock are of clinical importance [105]. The expression of USP2-4 peaks just before the onset of the dark phase [11,87,89]. Like BMAL1/CLOCK-controlled genes, an apparent diurnal rhythm of hepatic *Usp2-4* expression was severely inhibited in *Bmal1*KO mice, suggesting that USP2-4 is regulated by the liver clock. In addition to being affected by the circadian clock, *Usp2-4* mRNA levels in the liver were also controlled by nutritional input, although the effect was not seen in the white adipose tissue or skeletal muscle of mice [11]. In cultured hepatocytes, *Usp2-4* mRNA was increased by hydrocortisone and further augmented by glucagon, whereas insulin suppressed this hydrocortisone- and glucagon-induced *Usp2-4* expression [11]. Hepatic *Usp2-4* expression was also potentiated by PPARγ coactivator (PGC)-1α [82], which is a critical transcriptional coactivator for hepatic energy metabolic processes, such as gluconeogenesis, oxidative phosphorylation, and β-oxidation [106]. Collectively, the abundance of hepatic USP2-4 is regulated by the circadian rhythm and nutritional stimuli. Overexpression of *Usp2-4* in the liver increases blood glucose, insulin, and hepatic PEPCK levels, whereas *Usp2* knockdown in the liver elicits a hypoglycemic response and abolishes the daily oscillations in blood glucose levels in mice under night-feeding and day-feeding regimes [11]. Therefore, USP2-4 likely acts as a key regulator of glucose homeostasis by modulating liver function. Accordingly, the overexpression of *Usp2-4* induced through an adenovirus vector was found to exacerbate HFD–induced diabetes by causing insulin tolerance and glucose intolerance [11]. Knocking down hepatic *Usp2* in the liver of this model rescued the aberrant responses to insulin and glucose [11]. Given that USP2-4 positively regulates the expression of several glucocorticoid-regulated genes in the liver, USP2 appears to exacerbate glucose intolerance by stimulating glucocorticoid signaling [11]. Interestingly, USP2-4 potentiates the expression of the 11β-hydroxysteroid dehydrogenase 1 (HSD1) gene, which is an enzyme involved in the synthesis of cortisol, therefore implying that local glucocorticoids may participate in USP2-4–evoked gluconeogenesis [11]. Taken together with the fact that USP2-4 deubiquitinates C/EBPα, which potentially upregulates *HSD1* expression in hepatocytes [11], USP2-4 likely stabilizes C/EBPα to promote local glucocorticoid production, which results in the acceleration of gluconeogenesis.

The liver is also a primary organ for lipid metabolism; it synthesizes cholesterol from triglycerides and releases cholesterol into the circulatory system [107]. The liver also takes up low density lipoprotein (LDL) via the recognition of apolipoproteins by LDL receptors (LDLR) [108]. LDLR is polyubiquitinated by an E3-ubiqutinase–inducible degrader of LDLR (IDOL) and subjected to proteasome-dependent degradation [109]. Both USP2-1 and USP2-4 deubiquitinate and stabilize IDOL, suggesting that USP2 promotes LDLR degradation [110]. However, USP2 somewhat preserves LDLR stability and dampens IDOL-induced suppression of LDL uptake [110]. In agreement, USP2 increased surface LDLR levels and promoted LDL uptake in cultured hepatocytes [110]. An overexpression study implied that USP2 forms a tri-partite complex with IDOL and LDLR at the plasma membrane where it counteracts the IDOL-induced ubiquitination of LDLR and thereby prevents the digestion of LDLR by proteasomes [110]. Therefore, USP2 sustains LDLR-mediated LDL uptake and also stabilizes IDOL in hepatocytes. 

Macrophages are roughly classified as classically activated or alternative activated, categories that exhibit proinflammatory or tissue remodeling roles, respectively [111]. It is well established that a polarization toward classically activated macrophages aggravates insulin tolerance [112]. In terms of visceral adipose tissue, classically activated macrophages accumulate in the tissue of obese patients, leading to chronic inflammation in these tissues [113]. This chronic adipose tissue inflammation increases the circulating levels of “harmful” humoral factors, such as TNF-α and plasminogen activator inhibitor-1 (PAI-1), which results in insulin resistance or diabetes-associated tissue disorders [114,115,116]. We previously demonstrated that in human macrophage-like HL-60 cells, USP2-1 suppressed the expression of adipose tissue inflammation–related genes for *SERPINE1* PAI-1, adipocyte protein 2 (aP2), HMGA4high mobility group protein A2 (HMGA2), and matrix metalloproteases (MMPs) [9]. In agreement with these data, the expression of USP2-1 was significantly diminished in *ob/ob* mice, whereas *SERPINE1* (encoding PAI-1), *FABP4* (encoding aP2), and *HMGA2* were upregulated. Moreover, macrophage-selective *Usp2-1* transgenic mice showed a marked reduction in the number of adipose tissue macrophages in response to HFD feeding, indicating that macrophage USP2 likely facilitates macrophage infiltration into the visceral adipose tissues [9]. Furthermore, conditioned media from *USP2*-knockdown HL-60 cells stimulated the expression of adipogenic and inflammatory genes in 3T3-L1 cells, suggesting that macrophage USP2 controls adipose tissue remodeling [9]. Accordingly, the overexpression of *USP2-1* in macrophages restored insulin signaling in the liver and muscles in obese mice and reduced insulin tolerance [117]. All these observations demonstrate that USP2 in adipose tissue macrophages prevents the aggravation of obesity-induced diabetes. Regarding cellular signaling, *USP2* knockdown did not influence the phosphorylation of ERK, p38, JNK, p65 nuclear factor-κB (NF-κB), C/EBPα, signal transducer and activator of transcriptions (STATs), or the ubiquitination of mitogen-activated protein kinase phosphatases and NF-κB inhibitor (IκB) α in HL-60 cells [9]. The same cells also showed negligible changes in the nuclear abundance of PPARs and retinoic acid receptors (RXR), and no alteration in the binding activity of 17 transcription factors [9]. On the other hand, *USP2* knockdown potentiated histone H4 acetylation and histone H3 lysine 4 methylation of the *FABP4* and *HMGA2* loci [9]. Histone H4 acetylation causes conformational changes in the histone structure and facilitates the access of transcriptional factors to gene promoters [118]. Moreover, H3 lysine 4 methylation activates promoters or enhancers through interaction with transcriptional factor complexes, as well as with modifying enzymes for histone H4 acetylation [118,119]. Coinciding with these actions, *USP2* knockdown increased histone accessibility, suggesting that USP2 epigenetically impedes the expression of adipose inflammation-associated genes. Although the cellular contents of some deacetylases, histone methylases, and demethylases were constant in the *USP2* knockdown cells, further studies may uncover direct or indirect histone modifiers that are modulated by USP2 [9].

## 7. Nervous System

USP2 is abundantly expressed in neural cells in the intact mouse brain [120]. Correspondingly, an emerging body of evidence suggests significant roles for USP2 in the nervous system. As previously mentioned, USP2 modulates the molecular clock in the SCN by regulating BMAL1 stability, and presumably alters neuroendocrine activity [87]. Additionally, microarray data indicate that *Usp2* mRNA level is slightly, but significantly, increased in the hypothalamus and cerebral cortex of hypoglycemic mice [121]. Because the brain primarily utilizes glucose as its energy substrate [122], USP2 may therefore play a neuroprotective role against hypoglycemia.

Stress is known to evoke cognitive and emotional deficits [123]. In particular, stress-induced cognitive defects are attributable to the dysfunction of hippocampal channels or receptors [123,124]. Li et al. reported that, in rats, acute stress led to cognitive failure, accompanied by the downregulation of α-amino-3-hydroxy-5-methyl-4-isoxazolepropionic acid (AMPA) receptors in the hippocampal CA1 region [120], as well as significant decreases in the expression of PGC-1α, β-catenin, E4 promoter-binding protein 4 (E4BP4), and USP2. Because PGC1-α, β-catenin, and E4BP4 have been postulated as upstream regulators of USP2 [82], stress-induced attenuation of their expression may in turn attenuate hippocampal USP2. Retigabine, a drug that opens voltage-gated potassium channels, has been shown to abolish stress-induced cognitive defects; normalize the expression of PGC-1α, β-catenin, and USP2; and alleviate the increase in phosphorylated mammalian target of rapamycin (mTOR) and autophagy components [120]. Based on these findings, the authors speculated that USP2 deficiency impaired special memory retrieval via the aberrant expression of AMPA receptors and enhancement of mTOR-evoked autophagy [120]. 

Clinical surveys have indicated pivotal roles for USP2 in brain function. Blood RNA samples prepared from 13 patients with schizophrenia and 6 patients with bipolar disorder revealed that the *USP2* gene was negatively associated with scores on the Scale for the Assessment of Positive Symptoms; however, the genes for two ubiquitin conjugation enzymes, ubiquitin conjugating enzyme E2K, and seven in absentia homolog 2, were positively associated [125]. Presently, it is still unclear whether USP2 is aberrantly expressed in neural cells, or if certain mediators from blood cells may affect neural activity during the development of the central nervous system. Exome sequencing has identified *USP2* as one of the candidate genes exhibiting de novo and recessive variants, with substitutions in the 184^th^ amino acid of USP2-1 in patients with developmental delay, seizures, hypotonia, cryptorchidism, and club feet [126]. The affected site is required to bind MDM4, and is also predicted to influence subcellular localization and interaction with other proteins, such as MDM2 [126]. Therefore, USP2-mediated modulation of p53 signaling may be involved in normal brain development.

Studies using *Usp2*KO mice have generated insights into the physiological roles of USP2 in brain-mediated behaviors. For example, two groups reported that *Usp2*KO mice were more active when they were subjected to a wheel running test [86,87], whereas another group showed that the circadian free-running period and overall locomotor activity of the *Usp2*KO mice were not altered [90]. More recently, a preprint reported that *Usp2*KO mice exhibited an increased and more continuous active period compared to control mice [127]. *Usp2*KO mice also perform poorly in rotarod tests, novel object recognition tests, and acoustic startle reflex tests, thereby revealing defects in motor coordination, short-term recognition, and sensorimotor gating, respectively [127]. In addition, plus maze tests and novelty-suppressed feeding tests showed that the *Usp2*KO mice have decreased anxiety-like behavior [127]. The same study did not find any significant differences in appetite or spatial memory formation in *Usp2*KO mice, indicating that the roles of USP2 in the central nervous system are relatively specialized [127]. Future research may clarify other regulatory roles of USP2 in the brain, and may also uncover the signaling events underlying these phenomena.

## 8. Skeletal and Cardiac Muscles

The roles of USP2 in skeletal muscle differentiation have been evaluated in one of the first reports, which showed that USP2-1 and USP2-4 were differentially expressed in rat L6 myocytes. USP2-1 was temporarily increased in the early phase of differentiation, while the levels of USP2-4 gradually increased during differentiation [128]. Constitutive expression of *USP2-1* promoted the fusion of myoblasts along with the induction of myosin heavy chain, whereas *USP2-4* overexpression dampened myotube-like differentiation [128]. These results imply a potential role of USP2 in the development of muscle; however, knockout of the *Usp2* gene did not result in abnormal skeletal muscle phenotypes in mice, indicating that USP2 is dispensable for embryonic muscle development in vivo [129]. Further investigation is necessary to elucidate the specific roles of USP2 in muscle maintenance, including muscle wasting and muscle remodeling [130].

Tenderness is an economically important factor considered in the palatability of beef to consumers. Data from an mRNA-seq analysis of skeletal muscle in 24 Nellore cattle in Brazil that showed high or low shear force after 14 days of aging revealed molecules that are potentially involved in determining beef tenderness [131]. A complementary co-expression analysis using Partial Correlation with Information Theory indicated *USP2* mRNA was one of the most “differentially hubbed” transcripts, alongside growth factor receptor-bound protein 10 (*GBR10*), anoctamin 1 (*ANO1*), and transmembrane BAX inhibitor motif containing 4 (*TMBIM4*). Pathway analysis using the Kyoto Encyclopedia of Genes and Genomes suggested that USP2, GBR10, ANO1, and TMBIM4 are involved in the proteasome pathway [131]. Although a mechanistic explanation for the involvement of USP2 in beef tenderness is presently lacking, USP2 may affect the maturation and quality of muscle in cattle.

There is little knowledge regarding USP2-elicited molecular events in mature skeletal muscle, but we have previously explored mechanisms underlying these events in myoblasts. We found that knockout of the *Usp2* gene led to defects in proliferation and differentiation of mouse C2C12 myoblasts [132]. Furthermore, oxygen consumption and intracellular ATP were significantly reduced in *Usp2*-deficient myoblasts [132]. *Usp2*KO C2C12 myoblasts also exhibited an increase in fragmented mitochondria concomitant with an accumulation of ROS [132]. Treatment with ML364, a USP2 inhibitor, also elicited the accumulation of ROS-induced mitochondrial damage and a decrease in intracellular ATP. Together, these results indicate that USP2 is a prerequisite for the protection of mitochondria against ROS in myoblasts [132]. We then hypothesized that uncoupling protein 2 (UCP2) was responsible for the USP2-driven ROS removal. In support of this idea, we found that UCP2 was significantly decreased in *Usp2*KO C2C12 cells [132]. Further studies are required to elucidate how USP2 positively controls UCP2 expression.

Three publicly available mouse transcriptomics datasets indicate that *Usp2* is downregulated in the heart under overload pressures [133]. Transverse aortic construction (TAC), which mimics cardiac pressure overload, has clearly been shown to downregulate USP2 in cardiac muscle [133]. Infection of cardiac tissue with adeno-associated virus expressing *Usp2* considerably improved left ventricular (LV) contractile function, as evaluated by ejection fraction, fraction shortening, and LV internal dimensions; LV anterior wall thickness; and posterior wall thickness at end-diastole and end-systole [133]. In addition, *Usp2* overexpression suppressed cardiac hypertrophy, infiltration of CD68^+^ inflammatory macrophages, induction of proinflammatory cytokines, myocardial fibrosis, and increases in collagen mRNA after TAC [133]. Overexpression of cardiac *Usp2* also reduced oxidative stress, attenuated TAC-elicited induction of genes encoding NADPH oxidase (NOX)2, NOX4, and p22^phox^, downregulated TAC-induced phosphorylation of Akt, ERK, and Iκβ kinase, and mitigated the nuclear localization of NF-κB p65 [133]. Therefore, USP2 may attenuate cardiac remodeling during pressure overload by disrupting the signaling processes. 

The α1C subunit of the voltage-dependent L-type calcium channel (Ca_v_1.2) is responsible for calcium influx leading to calcium-induced calcium release from the endoplasmic reticulum in cardiac muscle [134]. Phosphorylation of Ca_v_1.2 by protein kinase A promotes the opening of L-type calcium channels in cardiac muscle; therefore, Ca_v_1.2 is a major target of the sympathetic nervous system in the heart [134]. An overexpression study using kidney cells demonstrated that USP2-4 decreased the surface expression of Ca_v_1.2 and attenuated Ca_v_ currents by suppression of ubiquitination of Cav1.2 and α2δ-1 subunits [135]. Because USP2-4 only associated with the α2δ-1 auxiliary subunit, but not the Ca_v_1.2 subunit, the negative effects of USP2-4 on surface Ca_v_1.2 is therefore likely attributable to the α2δ-1 subunit [135]. In other words, USP2-4 determines surface Ca_v_ availability via deubiquitination of the α2δ-1 subunit. Therefore, upregulation of USP2-4 by certain stimuli may alter the contraction of cardiac muscle.

## 9. Immune and Inflammatory Responses

The role of USP2 in immune and inflammatory signaling is still controversial. Discrepancies in findings may be caused by different experimental variables, such as cell types, stimulation types, duration of stimulation, inflammatory indexes, and sample species (Figure 3).

TNF-α, a canonical inflammatory cytokine, comprises an active complex with TNF receptor-1, TNF receptor–associated death domain protein (TRADD), RIP1, and TGF-β–activated kinase 1 (TAK1) [136,137,138]. This TNF-α-elicited complex stimulates the IκB kinase complex, which is followed by the phosphorylation-dependent polyubiquitination of IκBα [136,137,138]. Subsequently, IκBα is digested in the proteasome, resulting in the nuclear retention of NF-κB complexes and the induction of proinflammatory genes [136,137,138]. To date, a growing body of evidence indicates that USP2 modifies NF-κB activity via several mechanisms. Metzig and his colleagues performed siRNA screening to identify the USPs that participate in TNF-α-induced NF-κB activation, and reported that *USP2* siRNA attenuated ~50% and ~30% of κB site–driven luciferase reporter activity in HEK293 cells and HepG2 cells, respectively. *USP2* siRNA also inhibited the nuclear translocation of NF-κB p65 in HeLa cells, and suppressed the induction of transcripts for C-C motif ligand (CCL) 2, C-X-C motif ligand 2, interleukin (IL)-8, and IκBα [139]. These results indicate that USP2 mediates the induction of TNF-α-elicited proinflammatory cytokines by activating NF-κB.

TNF-α is reported to have a pivotal role in the pathogenesis of rheumatoid arthritis (RA) [140,141]. Akhtar et al. investigated the TNF-α signaling molecules regulated by miR-17, which is decreased in the serum, synovial fibroblasts, and synovial tissues of patients with RA and rats with adjuvant-induced arthritis. A bioinformatics analysis of a miR-17 gain-of-function study predicted that miR-17 may influence the expression of *TRAF2* and *BIRC3* (encoding cellular inhibitor of apoptosis protein 2 (cIAP2)). Like these proteins, *USP2* is also a miR-17–affected gene. The introduction of pre-miR-17 dramatically suppressed USP2 expression in TNF-α–pretreated synovial fibroblasts [142]. Because treatment with pre-miR-17 augmented the polyubiquitination of TRAF2 and cIAP1/2 in synovial fibroblasts, the authors speculated that USP2 may regulate the stability of these signaling molecules involved in the TNF signaling cascade.

TNF-α protein has also been reported be a direct target of USP2. Bacterial lipopolysaccharide (LPS), a toll-like receptor 4 ligand, induced miR-124 expression in mouse macrophage-like RAW264.7 cells [143]. In these cells, miR-124 knockdown potentiated the production of LPS-elicited TNF-α protein, while treatment with a miR-124 mimic ablated TNF-α synthesis [143]. Because miR-124 did not modulate TNF-α at the mRNA level in response to LPS stimulation, the authors concluded that miR-124 enhanced TNF-α production via a post-transcriptional mechanism. Indeed, the TargetScan database suggests that putative miR-124-binding sites are present on the 3’ untranslated region of *Usp2* transcripts [143]. Furthermore, USP2 knockdown in RAW264.7 cells shortened the half-life of the TNF-α protein, attenuated TNF-α release, and abolished the effects of miR-124 on TNF-α protein stability [143]. Therefore, USP2 stabilizes TNF-α in RAW264.7 macrophage-like cells.

Besides innate immunity and proinflammatory responses, NF-κB also contributes to adaptive immunity [144]. In T-lymphocytes, activation of the T-cell receptor (TCR) complex promotes the recruitment of caspase recruitment domain family member 11 (CARD11) to the lipid raft, and stimulates the formation of the CBM complex, which is a signalosome comprising CARD11, B-cell lymphoma 10 (BCL10), and mucosa-associated lymphoid translocation gene 1 (MALT1) [144]. Meanwhile, TRAF6, which is an E3-ubiquitin ligase, promotes K63-linked polyubiquitination of MALT1, BCL10, and IκB kinase (IKK)γ, and leads to the association of the CBM complex with the IKK complex [144]. TRAF6 also activates the TAK1/TAK1-binding proteins (TABs) complex, which phosphorylates IKKβ and in turn activates the IKK complex [144]. Eventually, the activated IKK complex induces the nuclear localization of NF-κB via proteasome-dependent IκBα-degradation [144]. So far, USP2-1 has been shown to associate with MALT1 and CARD11, in response to combinatory treatment using phorbol 12-myristate 13-acetate (PMA) and ionomycin to mimic TCR activation in Jurkat T cells [145]. USP2-1 is also constitutively bound to TRAF6 in these cells [145,146]. USP2 appears to mediate TCR signaling, because its knockdown suppressed TCR activation-provoked phosphorylation of IκBα and subsequent IL-2 production [145]. In agreement, USP2 modifies the ubiquitination of TRAF6, and thereby mediates its recruitment to MALT1 [145]. Given that USP2-1 digests the small ubiquitin-related modifier (SUMO) chain on TRAF6 [145], and that sumoylation interferes with protein–protein interactions, USP2 may facilitate the interaction between TRAF6 and CBM to trigger NF-κB activation in T-lymphocytes. Because *USP2* gene was transcriptionally activated in splenic B-lymphocytes in response to stimulation with IL-4, the role of USP2 in lymphocyte activation may not be restricted to T-lymphocytes, but may also encompass B-lymphocytes [147].

In contrast to the reports above, USP2 has also been shown to negatively regulate proinflammatory and immune responses. We previously demonstrated that stimulation with LPS led to the downregulation of *USP2* in the human macrophage cell line HL-60, mouse macrophage cell line J774, and mouse peritoneal macrophages [10]. *USP2* knockdown also potentiated the expression of 25 out of 104 cytokines in LPS-stimulated HL-60 cells, whereas the reintroduction of *USP2* isoforms to the *USP2* knockdown cells blocked the enhanced expression of these 25 cytokines [10]. Accordingly, isolated macrophages from macrophage-selective *Usp2-1* transgenic mice exhibited suppressed induction of genes for proinflammatory cytokines, such as TNF-α, IL-6, and CCL4, in response to LPS stimulation [10]. Taken together, USP2 represses the induction of proinflammatory cytokines in macrophages. The TRAF6/NF-κB signaling pathway does not appear to be a direct target of USP2, because knockdown or overexpression of *USP2* did not modify the polyubiquitination of TRAF6, or the levels of TRAF6, nuclear NF-κB, or cytoplasmic IκBα [10]. However, *USP2* knockdown caused a significant decrease in octamer binding protein (OCT)-1 and -2 in the nucleus [10]. The binding ratio of OCT-1 to OCT-2 to cytokine promoters was also significantly increased in *USP2*-knockdown HL-60 cells [10]. Further, USP2 deubiquitinated the K48- and K63-linked polyubiquitin chains on OCT-1, but not OCT-2 [10]. Hence, modulation of the ratio of the OCT proteins occupying the cytokine promoters underlies USP2-modified cytokine expression.

IL-1β and Sendai virus (SeV) also activate NF-κB via the TRAF6/NF-κB signaling cascade [146]. Overexpression of *USP2-1* drastically suppressed NF-κB activity in HEK293 cells in response to IL-1β treatment or SeV infection [146]. Conversely, treating *Usp2*KO cells with IL-1β or SeV elicited NF-κB hyperactivation, followed by robust induction of proinflammatory cytokines, such as TNF-α and IL-6 [146]. USP2-1 directly interacted with the RING finger and Zinc finger domains of TRAF6, and deubiquitinated the K63-linked polyubiquitin chain on TRAF6 [146]. Because USP2 deficiency leads to elevated TRAF6 levels in HEK293 cells independently of IL-1β or SeV stimulation, USP2 may therefore repress TRAF6/NF-κB signaling during the steady state.

Pathway-sensing pathogen-associated molecular patterns (PAMPs) are well conserved in vertebrates and invertebrates [148]. During bacterial infection, *Drosophila* macrophages exhibited activation of NF-κB–like proteins such as Relish, which led to the induction of antimicrobial peptides [149]. Overexpression of *Usp2* suppressed the expression of *Dpt* (encoding diptericin), an antimicrobial peptide, and simultaneously reduced survival rate in *Drosophila* [150]. Conversely, silencing the *Usp2* gene promoted the expression of *Dip* and *AttA* (encoding attacin A), another antimicrobial peptide, pre and post bacterial challenge [150]. The USP2 deficiency–elicited induction of antimicrobial peptide genes was dependent on Immunodeficiency (Imd), a *Drosophila* orthologue of RIP1, whose K48-linked polyubiquitination cleavage was catalyzed by USP2 in macrophage-like S2 cells [150]. Although K48-linked polyubiquitination generally triggers proteasome-dependent digestion of the ligated protein, the removal of the polyubiquitination chain from Imd instead accelerated its degradation in S2 cells [150]. Compared to Imd or USP2 alone, the Imd–USP2 complex was clearly associated with proteasome subunits, suggesting that USP2 likely conveys Imd to the proteasome [150]. Therefore, USP2 moderates excess activation of Relish-dependent immunity by promoting Imd degradation.

Interferon (IFN) plays a critical role in antiviral immunity, and is broadly classified into types I–III [151]. IFN type I (IFNα/β) and type III (IFNλ) activate receptor-associated Janus kinase (JAK) 1 and tyrosine kinase 2 (Tyk2), whereas type II IFN (IFNγ) promotes the activation of JAK1 and JAK2 [152]. JAK1/Tyk2 and JAK1/JAK2 phosphorylate STAT1/STAT2 heterodimers and STAT1 homodimers, respectively [152]. The phosphorylated STAT complexes translocate to the nucleus and bind to the IFN-stimulated response element or the IFNγ activation site to induce hundreds of antiviral IFN-stimulated genes [153]. The phosphorylation at tyrosine 701 (pY701) determines the stability of nuclear STAT, and the polyubiquitination and subsequent proteosome digestion of pY701-STAT1 occurs in the nucleus [154]. USP2-1 has been shown to directly deubiquitinate pY701-STAT1, which results in the accumulation of pY701-STAT1 in the nucleus [154]. Because IFNs accelerate the translocation of USP2-1 into the nucleus [154], the resulting induction of antiviral proteins by the stabilized STATs therefore confers viral resistance. Correspondingly, *USP2* knockdown decreased antiviral activity in embryonic kidney 293T cells, HepG2 cells, fibrosarcoma 2fTGH cells, and epidermoid KB cells against the vascular stomatitis virus (VSV); in human vascular endothelial cells against the dengue virus; and in lung cancer A549 cells against the influenza A virus (H1N1 strain) [154].

In addition to modulating the downstream signaling of the IFN receptors, USP2 has also been postulated to alter the production of IFN type I. Infection with SeV caused a small decrease in the levels of both USP2-1 and USP2-4 in HEK293 cells and macrophage-like THP-1 cells [155]. Moreover, USP2-4, but not USP2-1, attenuated the activation of *IFNβ* promoter after SeV infection, indicating a role for USP2 in suppressing viral immunity [155]. USP2-4 also inhibited the activation of the *IFNβ* promoter by modulating retinoic acid-inducible gene-I (RIG-I), melanoma differentiation-associated gene 5 (MDA5), mitochondrial antiviral signaling protein (MAVS), TANK binding kinase 1 (TBK1), and possibly activating interferon regulatory factor 3 (IRF3) [155]. USP2-4 removed the K63-linked polyubiquitin chain from TBK1, but did not alter the polyubiquitination of RIG-1 and MAVS [155]. Furthermore, USP2-4 repressed SeV-elicited TBK1 activation [155], suggesting that TBK1 is a direct target of USP2-4. USP2-4-mediated modulation of RIG-I/TBK1 signaling appears to influence antiviral activity because the level of *USP2-4* expression determined susceptibility to VSV infection [155]. In addition, USP2-4 also inhibited the activation of the Toll/IL-1R domain-containing adaptor, which induces IFN-β (TRIF) and cyclic GMP-AMP synthase (cGAS)/stimulator of interferon genes (STING) signaling, and attenuates *IFNB1* (encoding IFN-β) expression [155]. Given that both TRIF-mediated and STING-mediated pathways employ TBK1 to induce IFNβ, USP2-4 is therefore a common suppressor of antiviral signaling pathways.

The potential involvement of USP2 in the pathogenesis of nephritis has been previously documented. USP2 expression was found to be upregulated in exceptionally proliferative cells in glomerulonephritides in parallel with increased pathological severity [156]. Since mesangial cells undergo proliferation in various types of glomerulonephritides, USP2 is likely to be upregulated in mesangial cells during nephritides [156]. Concordantly, the expression of USP2 protein was potentiated in cultured mesangial cells by IL-1β, anti-thymocyte serum (ATS), and ATS with fresh human serum, indicating that inflammatory activation of mesangial cells elicits USP2 expression. The biological roles of USP2 in mesangial cells have also been investigated in a rat model. The intravenous injection of anti-thy1.1 antibody causes severe nephritis, increases proliferative cells, accumulates α-smooth muscle actin and collagen IV in glomeruli, and induces TGF-β, which stimulates fibrosis in inflamed regions [48]. Administration of a *USP2-1*–expressing plasmid into this rat nephritis model markedly abated the accumulation of proliferative cells and the deposition of extracellular matrix [48]. Because the expressed USP2-1 was primarily observed in the mesangium of the glomeruli, USP2-1 may suppress the hyperactivation of mesangial cells [48]. Overexpression of *USP2-1* also promoted the induction of renal decorin, which suppressed the induction of TGF-β1 and collagen IV in cultured mesangial cells after treatment with TNF-α, TGF-β1, or platelet derived growth factor-BB [48]. Therefore, the induction of decorin in mesangial cells appears to underlie the therapeutic effects of USP2-1. 

## 10. Male Genital Tract

USP2 is considered to have a vital role in spermatogenesis because it is abundant in late elongating spermatids [5]. Although *Usp2*KO mice exhibited a normal phenotype, including the testis, with no obvious changes in the number and morphology of epididymal spermatids or testicular spermatozoa, they displayed severe male subfertility that entailed abnormal aggregation of elongating spermatids in the testis and multinucleated bodies in the epididymis [129]. Correspondingly, the in vitro fertilization efficacy of *Usp2*KO mouse sperm was remarkably low when compared to wild-type mouse sperm [129]. Because the fertilization rate from intracytoplasmic sperm injection and the degree of acrosome reaction were comparable between sperm from *Usp2*KO and wild-type mice, the subfertility observed in *Usp2*KO mice may therefore be caused by defects in sperm motility. Although *Usp2*KO sperm exhibited normal hyperactive motility in Hanks medium M199, they were complete immotile in phosphate-buffered saline (PBS) [129]. Given that sperm isolated from wild-type mice maintained motility even in PBS, USP2 therefore enables sperm to sustain hyperactive motility with a minimal supply of nutrients and ions.

We have recently reported the effects of a macrophage-selective *Usp2*KO (ms*Usp2*KO) on male genital tract function. Deficiency of macrophage USP2 did not cause vital changes to the number, localization, or proportion of testicular macrophage subpopulations [157]. Likewise, there were negligible changes in the composition and localization of other testicular cells, including male gametes, Sertoli cells, and Leydig cells, with no obvious changes in testicular and circulating testosterone levels [157]. The loss of macrophage USP2 also did not affect the number, morphology, motility, or hyperactivation of freshly isolated sperm, but aberrant mitochondrial oxidative phosphorylation, capacitation, and hyperactivation were observed in freeze-thawed ms*Usp2*KO mouse sperm [157]. Administration of macrophage-derived granulocyte macrophage-colony stimulating factor (GM-CSF) potentiated total motility in the ms*Usp2*KO sperm by maintaining the mitochondrial ATP supply, but did not restore capacitation or hyperactivation [157]. Thus, future studies should elucidate the mechanisms underlying GM-CSF induction in testicular macrophages and investigate the intercellular and intracellular events in testis that involve macrophage USP2.

Recently, a study of a large cohort of patients with sex development disorders revealed that a mutation at the *USP2* locus [c.550G4A:p. (Gly184Arg)] was associated with undescended testes with hypoplastic scrotum [126,158]. However, the cause of the phenotypic differences observed between *Usp2*KO mice and human patients with mutations in the *USP2* gene has not yet been elucidated. 

## 11. Perspectives

As discussed in this review, USP2 is currently proposed to be a multifunctional DUB that plays crucial roles in various physiological and pathological events. However, a large body of evidence pertaining to its molecular function was obtained from cultured cells. To acquire accurate knowledge of the roles of USP2 at the level of the whole organism, the functions of USP2 should be individually reevaluated. Accordingly, there are several examples showing that in vivo models do not replicate the putative USP2 functions suggested by culture studies. For example, *Usp2*KO mice exhibit normal growth, including skeletal muscle development [129], whereas the overexpression of wild type and dominant-negative forms of USP2 strongly modified differentiation in cultured myoblasts [128]. Likewise, USP2 modified surface ENaC abundance via several mechanisms in cultured cells [93,94,95,96,97], but *Usp2*KO mice maintain a normal sodium balance and blood pressure [102]. From this perspective, further studies using *Usp2*KO mice or specific USP2 inhibitors may be useful for the functional validation of the roles of USP2 in vivo. Meanwhile, drug specificity should be taken into consideration, because many chemical inhibitors have unexpected molecular targets [159]; for example, a recent paper reported that a USP2 blocker also inhibits severe acute respiratory syndrome coronavirus 2 papain-like protease [160]. Considering that these inhibitors can affect enzymes with similar structures, chemical inhibitors of USP2 could perturb other endogenous proteins. To reconcile the differences in results obtained from cellular models and animal models, the molecular network of compensation for USP2 in *Usp2*KO mice should be examined using more comprehensive approaches, including transcriptomics and proteomics. In particular, there is a paucity of studies on other DUBs that share the physiological roles of USP2 in vivo.

Previous reports have identified possible targets of USP2, including transcriptional factors and their modulators [10,11,86,87,88,89], ion channels [93,95,96,97,135,161], hormone receptors [100,162], signal adaptor proteins [77,78,81,145,146,163], cytokines or cytokine receptors [52,66], and metabolic enzymes [20,26]. Although the modification of these factors may account for some of the USP2-elicted responses, it is presumed that not all USP2 targets have been discovered. For example, some reports indicate that USP2 potentially contributes to memory formation in the hippocampus and cerebral cortex [120]. The target of USP2 in this context has not yet been discerned, although mTOR and AMPA receptor 1 have been postulated [120]. Similarly, we demonstrated that USP2 in testicular macrophages conferred cryoprotection to sperm via GM-CSF production, but a mechanistic explanation for GM-CSF induction by USP2 is not clear at present [157]. The ectopic expression of USP2 has been shown to deubiquitinate an enormous number of proteins in cultured cells [46], and USP2 potentially modulates a far larger number of key molecules that determine physiological responses. Overexpression models of USP2 and the candidate proteins have generally been employed to evaluate USP2 target candidates [10,11,46,87,94]; however, given that overexpression often generates abnormal cellular distributions, the interaction between USP2 and the target candidates require further assessment under physiological conditions. Moreover, excess USP2 may cause nonspecific off-target deubiquitination in cells and animals. Therefore, knockout or knockdown experiments, alongside intracellular co-localization studies in a physiological setting, can substantiate hypotheses on the interaction between USP2 and the target(s) of interest. 

Among the various types of polyubiquitin chains, the structure, interactors, and physiological significance of K48-linked and K63-linked polyubiquitination have been extensively documented [164,165,166]. In general, the K48-linked polyubiquitination moiety is known to be a canonical signal for proteasome-dependent degradation, while K63-linked polyubiquitin chains have been shown to modulate protein–protein interactions [165,166]. Additionally, recent advances in proteomics have revealed a large variety of protein ubiquitination, including linear ubiquitination with other lysine residues, branched ubiquitination, and monoubiquitination [165,167,168]. Diverse ubiquitin modifications confer distinct conformations to the target protein, which triggers various consequences in cells. Thus, the association of the ubiquitin (or ubiquitin-like molecule) chain on proteins with biological outcomes generates a so-called “ubiquitin code”, which facilitates our understanding of their biological relevance [165,169]. To date, few papers have profiled ubiquitin chains at a whole-genome scale in *USP2*-engineered cells. Therefore, approaches using “ubiquitinomics” may shed new light on the biological function of USP2 [170,171,172]. 

As described in this review, USP2 has pathophysiological roles in various tissues. Since the roles of USP2 are closely associated with pandemic disorders, including cancer and metabolic diseases, clinical and preclinical studies targeting USP2-associated signaling are desirable. In particular, since USP2 modulates the stability of critical tumor-associated proteins such as cyclin D1, Mdm2, and FASN, great efforts have been made to establish a USP2-targeting anticancer drug. ML364 was reported as a small molecule inhibitor for USP2 and has been proven to evoke cell cycle arrest, cyclin D1 degradation, and inhibition of homologous recombination-mediated DNA repair [42]. Additionally, several studies have reported other chemical inhibitors for USP2, such as isoquinoline-1,3-dione-based compounds [172], chalcone-based compounds [173], 5-(2-thienyl)-3-isoxazoles [43], and a lithocholic acid derivative [40]. Furthermore, other reports have demonstrated that previously established anti-cancer drugs inhibit USP2 activity. The *ortho*-quinone natural product β-lapachone, which has been subjected to phase II clinical trials for cancer therapy, was shown to target USP2, resulting in induction of apoptosis in the DU-145 prostate cancer cell line [174]. Moreover, a popular leukemia drug, 6-thioguanine, was shown to be a noncompetitive inhibitor for human USP2 [175]. In addition to evaluation of these drugs for anti-tumor therapy, these drugs could also be tested as treatments for other diseases where the pathology includes contributions from USP2.

The USP2 isoforms are expressed in a wide variety of tissues, albeit with expressional diversity between organs [9,12,13]. This suggests that USP2 has a common fundamental role across different cell types. Generalization of the properties of USP2 may facilitate the discrimination of USP2 from other DUBs. In the context of enzymatic activity, USP2 is known to be a relatively highly active DUB, similar to USP21 [176,177]. However, USP2 can also be characterized in terms of biological processes. Presently, several reports indicate that USP2 is closely associated with energy metabolism, although its targets are organ-specific. As mentioned in Section 6, USP2 modifies the expression of genes involved in lipid and glucose metabolism, such as FASN [18], IDOL [110], PEPCK, and G-6-Pase [11]. Conversely, hypoglycemia [121], adiponectin [41], and PGC-1α [82], all of which are highly linked to cellular energy metabolism, control USP2 expression. Moreover, the aberrant expression of USP2 in adipose tissue macrophages [9,117] and myoblasts [132] causes metabolic defects in vivo and in vitro, respectively. The metabolic effects of USP2 are also observed in sperm, which are regulated by USP2 in testicular macrophages. Therefore, USP2 may regulate various cellular functions by modulating energy metabolism. In particular, a number of recent papers demonstrate that energy metabolism states predominantly influence immunological function, suggesting that USP2 controls immune and/or proinflammatory responses of lymphocytes or macrophages by altering cellular metabolism [178,179,180,181,182]. A review of the molecular events involving USP2 may facilitate the characterization of its common functions and enable the understanding of its significance at an individual level. 

## 12. Short Summary and Conclusions

The findings detailed in this article are summarized in Table 1. USP2, mainly USP2-1, participates in tumorigenesis in various cancers through the potentiation of the cell cycle, mitosis, lipogenesis, metastasis, EMT, and anti-oxidation, while inhibiting p53-mediated tumor death. USP2 also controls TNF-elicited apoptosis as well as TNF resistance. Additionally, cFLIP-regulated autophagy is also regulated by USP2. USP2, especially USP2-4, is a core component of the biological clock, and contributes to the circadian rhythm in the liver and SCN by deubiquitinating BMAL1 and PER1. In the intestine, USP2-4 determines rhythmic calcium absorption, resulting in the maintenance of calcium homeostasis. Additionally, in vitro data demonstrate the crucial roles of USP2-4 in ENaC and MR expression, suggesting USP2 controls sodium adsorption, although *Usp2*KO mice presented no obvious phenotypic changes, including blood pressure. Accumulating evidence indicates that USP2 is a determinant for energy metabolism in hepatocytes, myoblasts, and cancerous cells. Moreover, previous reports also suggest significant roles of USP2 in the intercommunications between energy-competent organs such as liver, skeletal muscle and adipose tissue. Thus, aberrant expression of USP2 is considered to provoke metabolic diseases, such as type 2 diabetes and atherosclerosis. USP2 also maintains neural activity in the brain, and is postulated to be involved in special memory retrieval, motor coordination, short-term recognition, sensorimotor gating, and anxiety-like behavior. In terms of skeletal and cardiac muscles, USP2 is considered to modify differentiation, contraction, and remodeling after pressure overloads, although *Usp2*KO mice displayed no obvious phenotypic changes under normal conditions. At the present time, the function of USP2 in immunoregulation is controversial. Some reports have demonstrated that USP2 stimulates the production of cytokines and anti-viral proteins through NF-κB and STAT1-dependent mechanisms. In contrast, USP2 has also been suggested to attenuate the production of cytokines and microbicide peptides by perturbing the OCT1/2, NF-κB, and IRF3 pathways. In agreement with the observation that USP2 is most abundantly expressed in testis, USP2 deficiency caused severe male sterility due to defects in sperm motility. This malfunction of sperm is accounted for by the lack of USP2 in sperm and male genital tract macrophages. Although previous efforts have clarified a wide variety of roles played by USP2, the detailed molecular mechanisms underlying these phenomena remain to be uncovered in further studies. Given that there are several discrepancies between experimental models, the reevaluation of data by modern sophisticated and comprehensive techniques is also required. 

## Figures and Tables

**Figure 1 ijms-22-01209-f001:**
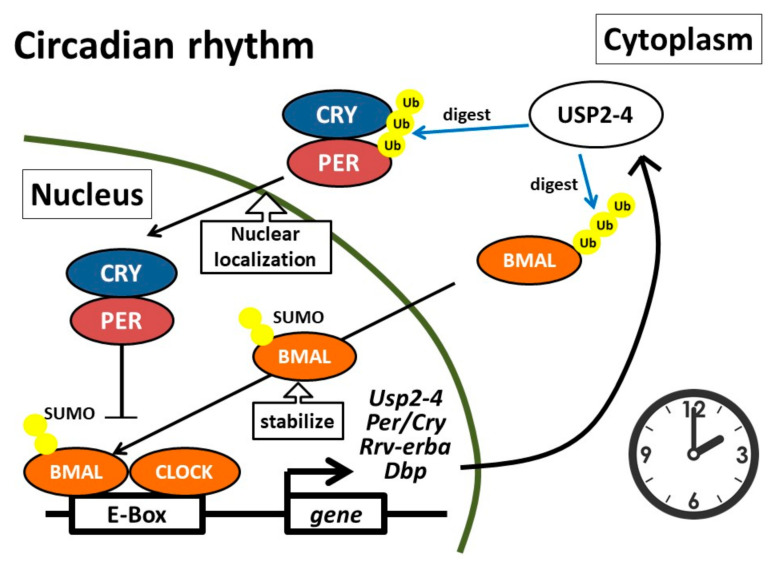
Roles of USP2-4 as a molecular clock regulator. USP2 is one of the known CLOCK/brain and muscle Amt-like protein 1 (BMAL1)-regulated molecules. One of the USP2 variants, USP2-4, digests the polyubiquitin chain on BMAL1, and subsequently promotes the accumulation of sumoylated BMAL-1 in the nucleus. Nuclear localization of BMAL1 leads to BMAL1/CLOCK-elicited gene expression. USP2-4 also digests the polyubiquitin chain on PER which results in prolonged nuclear localization of the PER1/cryptochromes (CRY) complex. CLOCK, Clock circadian regulator; PER, period.

**Figure 2 ijms-22-01209-f002:**
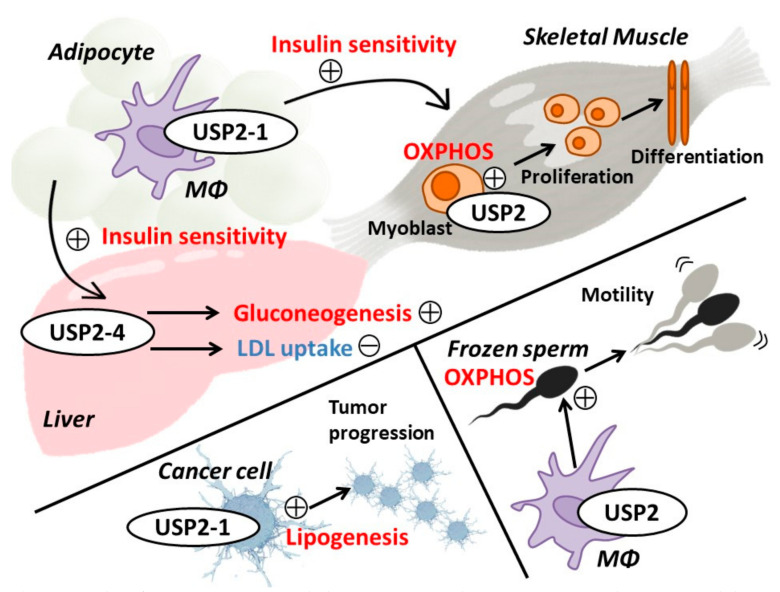
Roles of USP2 in energy metabolism. USP2-1 in adipose tissue macrophages (Mφ) inhibits chronic inflammation in adipose tissue, which restores insulin sensitivity in skeletal muscle in obese individuals. In the liver, USP2-4 promotes gluconeogenesis and increases low density lipoprotein (LDL) uptake by upregulating the LDL receptor. In myoblasts, USP2 maintains oxidative phosphorylation (OXPHOS) and ATP supply, which may contribute to myoblast proliferation and myotube differentiation. In various cancer cells, USP2-1 is a critical factor for the induction of fatty acid synthase, which participates in tumor progression. In the male genital tract, macrophage USP2 preserves OXPHOS activity in frozen sperm. Thus, deficiency of macrophage USP2 leads to impaired sperm motility.

**Figure 3 ijms-22-01209-f003:**
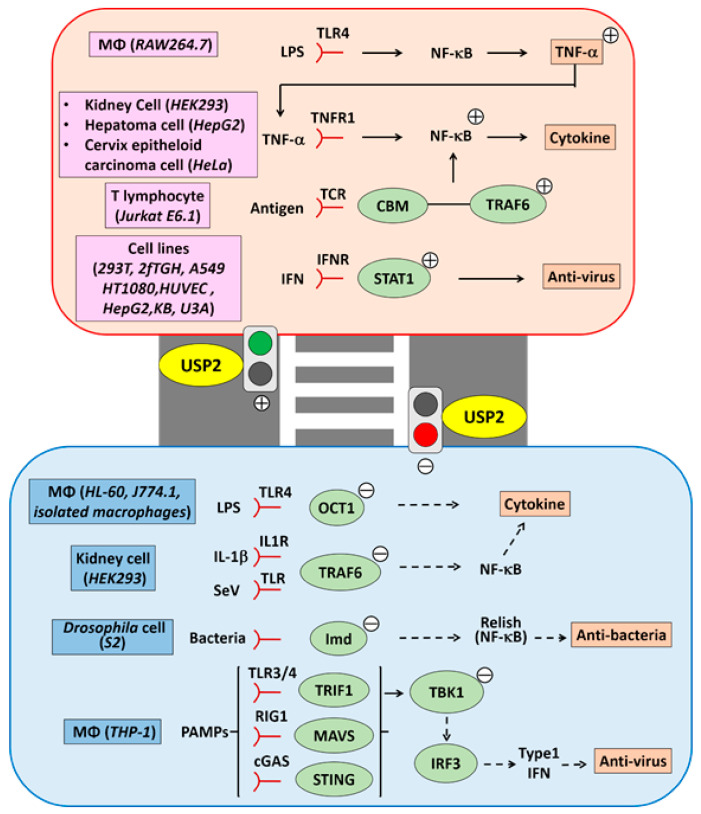
Roles of USP2 in immune and inflammatory responses. Previous reports indicate that USP2 exerts both positive and negative effects on immune and inflammatory responses. USP2 stabilizes TNF-α and the TNF-α–elicited signaling complex, leading to activation of NF-κB. USP2 also potentiates NF-κB activation in T-lymphocytes via the interaction between the CBM complex and TRAF6. Moreover, USP2 stimulates nuclear accumulation of STATs, which promotes antiviral activity in cells. In contrast, there are papers demonstrating that USP2 modifies OCT1, TRAF6, and Imd (*Drosophila* orthologue of RIP1) to suppress the expression of proinflammatory cytokines and anti-bacterial peptides. Additionally, USP2 modulates TBK1 to attenuate signaling along the RIGI, cGAS/STING, and TRIF pathways. USP2-elicited attenuation of TBK1 also lowered IRF3 activation, resulting in decreased IFNβ production.

**Table 1 ijms-22-01209-t001:** Pathophysiological roles of human and mouse USP2.

USP2 Species	USP2 Isoform	USP2-Affected Cell/Tissue/Animal	Experimental Manipulation	Direct Target	Function of USP2	Reference
Human	USP2-1	Cell line (Bladder cancer)	OE, KD	Cyclin A1	Proliferation (+), Invasion (+), Migration (+), Chemoresistance (+)	[24]
Human	USP2-1	Cell line (Breast cancer)	KD, Blocker	TWIST	EMT (+), Proliferation (+), Chemoresistance (+), Self-renewal (+)	[25]
Human	USP2-1	Cell line (Prostate cancer, Hepatoma)	OE, KD	FASN	Apoptosis (-), Lipogenesis (+)	[18,19]
Human	USP2-1	Cell line (Osteosarcoma, Lung carcinoma Embryonal carcinoma T-cell lymphoma)	OE, KD	MDM2	p53 signaling (-), Apoptosis (-), Chemo resistance (+)	[31,32]
Human	USP2-1	Cell line (Lung carcinoma Embryonal carcinoma)	OE, KD	MDMX	Chemo resistance (+)	[34]
Human	USP2-1	Cell line (Glioblastoma)	OE, KD	MDM4	p53 signaling (-), Apoptosis (-)	[35]
Human	USP2	Cell line (Hepatoma, Breast cancer)	KD	-	p53 expression (+)	[36]
Human	USP2-1	Cell line (Osteosarcoma, Kidney cells, Colorectal carcinoma, Breast cancer, Prostate cancer)	OE, KD	Cyclin D1	Cell cycle (+)	[39]
Human	USP2	Cell line (Hepatoma, Breast cancer)	OE, KD	Cyclin D1	Cell cycle (+)	[41]
Human	USP2-1	Cell line (Pancreatic carcinoma, Colorectal carcinoma, Kidney carcinoma)	OE, KD	Aurora-A	Proliferation (+), Mitosis (+)	[46]
Human Mouse	USP2-1	Mouse, Cell line (Cervix epithelioid carcinoma, Hepatoma, Colorectal carcinoma, Kidney cells),	OE, KD, KOBlocker	TGF-β receptor	TGF-β signaling (+), Metastasis (+), Tumor growth (+), Mortality (-)	[52]
Human	USP2-1	Cell line (Kidney cell)	OE, KD, Blocker	β-catenin	Wnt/β-catenin signaling (+)	[58]
Human	USP2	Cell line (Breast cancer), Mouse	OE, KD Blocker	ErbB2	Tumor growth (+), Proliferation (+), Cell cycle (+)	[62]
Human	USP2-1	Cell line (Prostate cancer)	OE, KD	ACDase	Proliferation (+)?	[70]
Human	USP2	Cell line (Kidney carcinoma)	OE	Unidentified	Proliferation (-), Migration (-), Invasion (-)	[74]
Human	USP2-2	Cell line (Cervix epithelioid carcinoma, Kidney cells)	OE	MDM2?	Apoptosis (+)	[7]
Human	USP2-1	Cell line (Prostate epithelial cells, Prostate cancer, Colon cancer, Breast cancer, Sarcoma, Fibroblasts) Mouse	OE, KD	Unidentified	Apoptosis (-), Cell growth (+), p53 signaling(-)	[75]
Human	USP2-1	Cell line (Hepatoma)	OE	ITCH?	Apoptosis (+)	[81]
Mouse	USP2-2	Tissue (Liver), Hepatocyte	OE, KD	ITCH	Apoptosis (+), TNF-resistance (-)	[13]
Human	USP2-1	Cell line (Cervix epithelioid carcinoma, Breast cancer, Kidney cell)	OE, KD	TRAF2, RIP1	Apoptosis (+) or (-), TNF-signaling (+), NF-κB activation (-), p38 signaling (-), JNK signaling (-)	[77,78]
Human	USP2-2	Cell line (Breast cancer)	OE, KD	TRAF2, RIP1	Apoptosis (+)	[78]
Human	USP2-1 USP2-2 USP2-3 USP2-4	Cell line (Kidney cell),	OE	Unidentified	Apoptosis (+)	[6]
Mouse	USP2-1 USP2-4	Mouse, Tissue (Liver) Fibroblast, Cell line (Kidney cell)	OE, KO	PER1	Control of circadian period, Circadian gene expression (+)	[86]
Mouse	USP2-4	Mouse, Tissue (SCN), Cell line (Kidney cell) Fibroblast	OE, KO	BMAL1	Control of circadian period, Circadian gene expression (+)	[87,88],
Mouse	USP2-4	Mouse, Fruit fly, Cell line (Kidney cell)	OE, KD, KO	NHERF4, Clathrin heavy chain	Calcium absorption (+), Sodium balance (n.i.)	[90,102]
Mouse	USP2-4	Cell line (Kidney cell)	OE	Ca_v_1.2, α2δ-1 subunit	Calcium uptake (-), Surface calcium channel (-)	[135]
Mouse	USP2-4	Tissue (Oocyte), Cell line (Kidney cell)	OE	ENaC, Nedd4-2	Sodium uptake (+) Surface ENaC expression (+) ENaC activation (+)	[93,94,95]
No information	USP2-4	Cell line (Kidney cell)	OE	Unidentified	Surface ENaC expression (+)	[96]
No information	USP2-4	Cell line (Kidney cell)	OE, KD	MR	Aldosterone signaling (-)	[100]
Mouse	USP2-4	Tissue (Liver), Cell line (Kidney cell)	OE, KD	C/EBPα	Gluconeogenesis (+) Glucose sensitivity (-) Insulin signaling (-) Glucocorticoid signaling (+)	[11]
Human	USP2-1 USP2-4	Fibroblast, Cell line (Kidney cell, Hepatoma, Epidermoid carcinoma, Cervix epithelioid carcinoma)	OE, KD	IDOL	LDL uptake (+)	[110]
Human, Mouse	USP2-1	Mouse, Tissue (Adipose tissue, Liver, Skeletal muscle), Cell line (Macrophage, Myocyte, Adipocyte)	OE, KD	Unidentified	Inflammation (-), Cytokine production (-), Insulin signaling (+) Adipocity (-) Histone modification (-)	[9,117]
Mouse	USP2	Tissue (Hippocampus)	Down-regulation by stress	mTOR?, AMPA receptor?	Autophagy (-)?, Spatial memory (+)?	[120]
Human	USP2-1	Human	Mutation	Unidentified	Development (+), Seizure (-), Muscle strength (+), Reproduction (+)	[126]
Mouse	USP2	Mouse	KO	Unidentified	Locomotion (+) or (n.i.) Motor coordination (+) Recognition (+) Sensory response (+) Anxiety (+)	[86,87,127]
Rat	USP2-1 USP2-4	Cell line (Myoblast)	OE, DN	Unidentified	Differentiation (+, USP2-1; -, USP2-4)	[128]
Mouse	USP2	Cell line (Myoblast)	KO, Blocker	UCP2?	Oxidative stress (-) ATP production (+) Proliferation (+) Differentiation (+)	[132]
No information	USP2	Tissue (Heart)	OE	Unidentified	Cardiac function (+), Fibrosis (-), Inflammation (-), Cytokine production (-), Oxidative stress (-), Akt signaling (+), NF-κB signaling (+), ERK signaling (+)	[133]
Human	USP2	Cell line (Kidney cell, Hepatoma, Cervix epithelioid carcinoma)	KD	Unidentified	NF-κB signaling (+), Cytokine production (+)	[139]
Mouse	USP2	Cell line (Macrophage)	KD	TNF-α?	Cytokine production (+)	[143]
Human	USP2-1	Cell line (Kidney cell, T cell)	OE, KD	MALT1, CARD11, TRAF6	TCR signaling (+), NF-κB signaling (+), Cytokine production (+)	[145]
Human, Mouse	USP2-1, USP2-4	Macrophage, Cell line (Macrophage)	OE, KD	OCT1	OCT1/2 signaling (-) Cytokine production (-)	[10]
Human	USP2-1	Cell line (Colorectal carcinoma, Kidney cell)	OE, KD, KO	TRAF6	NF-κB signaling (-), Cytokine production (-)	[146]
Fruit fly	USP2	Fruit fly, Cell line (Macrophage)	OE, KD	Imd	Antimicrobial activity (-), NF-κB signaling (-)	[150]
Human	USP2-1	Vascular endothelial cell, Cell line (Kidney cell, Fibrosarcoma, Lung carcinoma, Hepatoma, Epithelial carcinoma, Sarcoma)	OE, KD	STAT1	Antiviral activity (+), IFN signaling (+)	[154]
Human	USP2-4	Cell line (Kidney cell)	OE, KD	TBK1	Antiviral activity (-), TRIF signaling (-), STING signaling (-), IFNβ signaling (-), Cytokine production (-)	[155]
No information	USP2-1	Tissue (Kidney), Mesangial cell	OE	Unidentified	Inflammation (-), Fibrosis (-), Mesangial cell activation (-)	[48]
Mouse	USP2	Tissue (Testis), Sperm	KO	Unidentified	Sperm motility (+), Sperm capacitation (+), Spermatogenesis (+) Fertilization (+),	[129]
Mouse	USP2	Tissue (Testis), Sperm, Macrophage	KO	Unidentified	Sperm motility (+), Sperm capacitation (+), Fertilization (+), Cytokine production (+)	[157]

USP2 species, USP2 isoforms, USP2-affected materials, experimental manipulations, direct targets of USP2, biological function of USP2, and references are shown. OE, overexpression; KO, knockout; KD, knockdown; DN, dominant negative; (+), potentiation; (-), attenuation; (n.i.), not influenced. Question marks represent cases showing no direct evidence.

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
