# Peer review of "USP2-Related Cellular Signaling and Consequent Pathophysiological Outcomes"

_ijms, 2021, doi:10.3390/ijms22031209_

Round 1

Reviewer 1 Report

In this review manuscript the authors have undertaken a thorough literature review detailing studies assessing the de-ubiquitinating enzyme USP2. Due to the minute detail, the document could well qualify as a chapter or several chapters in a book, or alternatively several reviews highlighting different aspects of the deubiquitinase. Such a division would also allow for more detail to each part.

From reading the manuscript, the striking difference in effects reported for the enzyme emerges, whether over-expressed in cell lines and perhaps mice, or knocked out. This discrepancy is also mentioned in the final perspective section, which fairly well wrap up the review. The manuscript is well written, but perhaps with too much detail. Less room should perhaps have been given publications with lower impact. Please find enclosed a few points and suggestions.

Minor concern: The information of the different human and mouse isoforms of USP2, including their specific roles, could be presented in a table.

Abstract

Mention role of USP2 in metabolism.

The authors might consider mentioning discrepancy between experiments performed using in vitro and in vivo.

Tumorigenesis section

Line 90. Add “on the other hand” to the sentence: “USP2 has [on the other hand] also been shown to increase p53..”

Noncancerous proliferation section

Line 191-201. In this section there is only very limited evidence from one in vitro study that DIM inhibits USP2. The section could therefore be shortened (or omitted).

Apoptosis and autophagy section

Line 215. Is “proapoptotic” correct here? Perhaps the sentence should be rephrased.

Line 222-223. The sentence “Given the anti-apoptotic role of p53.. “ does not seem to make sense.

Circadian clock section

Line 315-317. Information in the sentence “Although Usp2KO mice..” could be better explained, including the statement within the parenthesis.

Line 355-358. It could be mentioned that the drosophila study did not find circadian changes.

Renal system section

Line 375. Describe in what situations Nedd4-2 polyubiquitinates ENaC.

Line 388. Briefly describe function of MRs. Next sentence: Describe what is phosphorylated.

Line 392. The authors expect that the readers have prior knowledge of function of aldosterone.

Energy metabolism and metabolic disorders section

Line 430: Define the term “phase-shifted”.

On line 434, hydrocorticosterone is mentioned, on line 435, hydrocortisone

Line 468. Find a better phrase than “bad”

Line 470 - 482. Associated with information provided in this section it could be mentioned that inflammatory macrophages are generally considered part of obese/diabetic pathology (see eg. PMID: 32140136).

Line 489. The authors could explain what is the consequence of histone H4 acetylation and histone H3 lysine 4 methylation.

Nervous system section

Line 540. Explain the reward and punishment systems is and why it is expected to be controlled by USP2.

Skeletal and cardiac muscle section

Line 589. Provide example of proinflammatory stimulation that activate USP2 in cardiac tissue.

Immune and inflammatory responses section

Figure 3. Whether cell lines have been assessed (MQ and T cells) this could be indicated in the figure.

Line 647. Add RAW264.7 to the sentence: “…stabilizes TNF- α in [RAW264.7] macrophages.”

Male genital tract section

Line 781 and 783. “Usp2KO mouse sperm” could be rephrased as the USP2 deficiency was limited to myeloid cells in the study.

Perspective section.

- The authors have stated that USP2 have a significant role in tumour-development and this is mentioned in the abstract. The authors could include a sentence, mentioning this.

- Whether overexpression of USP2 could result in unspecific off-target deubiquitination (also in mice) is something the authors could consider.

Author Response

Reviewer#1

I greatly appreciate the reviewer’s comments, and have revised the manuscript accordingly. My responses to the comment are provided below:

Comment 1: The manuscript is well written, but perhaps with too much detail. Less room should perhaps have been given publications with lower impact. Please find enclosed a few points and suggestions.

 Our Response: Thank you for your insightful comments. In this version, we have deleted the “Noncancerous” section (see Response to Comment 5), and also omitted sentences describing some other details (see Responses to Comments 8, 17, and 18).

Comment 2: The information of the different human and mouse isoforms of USP2, including their specific roles, could be presented in a table.

 Our Response: In regards to this point we have added Table 1, which shows the pathophysiological roles of human and mouse isoforms of USP2.

Comment 3 (Abstract): Mention role of USP2 in metabolism: The authors might consider mentioning discrepancy between experiments performed using in vitro and in vivo.

Our Response: We now briefly mention the discrepancies in the data obtained from in vitro and in vivo models as follows: “Additionally, we describe phenotypic differences found in the in vitro and in vivo experimental models.” (Lines 22-23).

Comment 4 (Tumorigenesis section): Line 90. Add “on the other hand” to the sentence: “USP2 has [on the other hand] also been shown to increase p53..”

 Our Response: According to this reviewer’s comment, we have revised the sentence as follows:

“On the other hand, USP2 has also been shown to increase p53 in a hepatoma cell line (HepG2) and a breast cancer cell line (MCF7) after leptin stimulation [35]. (Line 95)

Comment 5 (Noncancerous proliferation section): Line 191-201. In this section there is only very limited evidence from one in vitro study that DIM inhibits USP2. The section could therefore be shortened (or omitted).

Our Response: We completely agree with the reviewer’s idea that this paragraph only provides limited knowledge to the readers. Thus, we have deleted this paragraph from the present version.

Comment 6 (Apoptosis and autophagy section): Line 215. Is “proapoptotic” correct here? Perhaps the sentence should be rephrased. Line 222-223. The sentence “Given the anti-apoptotic role of p53.“ does not seem to make sense.

Our Response: As the reviewer pointed out, we made careless mistakes during the preparation process of the manuscript. We have carefully fixed this in the present version:

“In sharp contrast, Priolo et al. demonstrated that USP2-1 siRNA increased the apoptosis rate in the prostate cancer cell lines LNCaP (androgen dependent, wild type p53) and DU145 (androgen independent, mutant p53), but not PC-3 (androgen independent, p53-null), suggesting an anti-apoptotic role of USP2-1 [75].”(Lines 191-194)

“Given the proapoptotic properties of p53, it can be concluded that USP2-1 abates apoptosis by stabilizing MDM2, MDMX, MDM4, and FASN.”(Lines 200-201)

Comment 7 (Circadian clock section): Line 315-317. Information in the sentence “Although Usp2KO mice.” could be better explained, including the statement within the parenthesis.

Our Response: In the current version, we have revised the sentence to be more precise and have added further details as follows:

“Mammals can adjust circadian activities in response to irradiative light strength. The role of USP2 in circadian sensitivity to light has been also evaluated using Usp2KO mouse models. When Usp2KO mice were placed in constant light with low (but not high) irradiance after 12-hour light/dark conditions, the mice displayed a remarkable delay in the onset of the active phase [87]. Since mice increase or decrease their rhythmic activity in response to constant light with lower or higher irradiance, the delayed onset of the active phase in Usp2KO mice suggests that USP2 deficiency elevates sensitivity to low irradiative light. To support this, exposure to low irradiative light also caused a delay in the onset of the active phase after mice were housed in constant darkness [87].” (Lines 293-300)

Comment 8 (Circadian clock section): Line 355-358. It could be mentioned that the drosophila study did not find circadian changes.

Our Responses: As the reviewer pointed out, the Drosophila data does not provide evidence that USP2 is involved in circadian control. Thus, we have omitted this data from the current version of our manuscript.

Comment 9 (Renal system section): Line 375. Describe in what situations Nedd4-2 polyubiquitinates ENaC.

Our Response: In this version, we have added an explanation regarding the regulation of Nedd4-2 activity as follows: “Phosphorylation of Nedd4-2 is catalyzed by serum and glucocorticoid-regulated kinase, which is induced by steroid hormones such as aldosterone, an endogenous mineralocorticoid [93].” (Lines 356-358)

Comment 10 (Renal system section): Line 388. Briefly describe function of MRs. Next sentence: Describe what is phosphorylated.

Our Response: In accordance with the reviewer’s comment, we now briefly explain the function of MRs. Additionally, we also describe the phosphorylation of MRs after aldosterone stimulation in an ERK-dependent manner.

“In addition to surface ENaC expression, USP2-4 also determines the stability of mineralocorticoid receptors (MRs). MRs are receptors for aldosterone and induce expression of ENaC in epithelial cells leading to the reabsorption of sodium and water, and the consequent regulation of body fluid volume and blood pressure [99]. In addition to binding to MRs, aldosterone stimulates the phosphorylation of MRs in an extracellular signal regulate kinase (ERK)-dependent fashion [100]. Phosphorylated MRs are subsequently monoubiquitinated, which allows interaction with tumor suppressor gene (TSG) 101. Interaction with TSG101 elevates the stability of MRs [100].” (Lines 369-375)

Comment 11 (Renal system section): Line 392. The authors expect that the readers have prior knowledge of function of aldosterone.

Our Response: As mentioned above, we now briefly describe the basic function of aldosterone and its receptor (MR) for readers of IJMS coming from a broad range of backgrounds.

“MRs are receptors for aldosterone and induce expression of ENaC in epithelial cells leading to the reabsorption of sodium and water, and the consequent regulation of body fluid volume and blood pressure [99].”(Lines 370-372)

Comment 12 (Energy metabolism and metabolic disorders section): Line 430: Define the term “phase-shifted”.

Our Response: Although the data in the reference showed a slight shift of the diurnal expressional peak of USP2-4 to the dark phase in Bmal1KO mice, the authors did not mention this. Therefore, we have omitted the “phase-shifted” description from this version of our manuscript. 

Comment 13 (Energy metabolism and metabolic disorders section): On line 434, hydrocorticosterone is mentioned, on line 435, hydrocortisone.

Our Response: As the reviewer pointed out, we made a careless mistake. We have fixed this by changing “hydrocorticosterone” to “hydrocortisone”. (Line 420)

Comment 14 (Energy metabolism and metabolic disorders section): Line 468. Find a better phrase than “bad”

Our Response: We have replaced “bad” with “harmful” in this version. (Line 458)

Comment 15 (Energy metabolism and metabolic disorders section): Line 470 - 482. Associated with information provided in this section it could be mentioned that inflammatory macrophages are generally considered part of obese/diabetic pathology (see eg. PMID: 32140136).

Our Response: Although in the previous version we mentioned that classically activated (also called inflammatory or M1) macrophages contribute to adipose tissue inflammation, which has consequences for diabetes pathology, we have now added a new sentence indicating that macrophage polarization is a determinant for insulin tolerance.

“Macrophages are roughly classified as classically activated or alternative activated, categories that exhibit proinflammatory or tissue remodeling roles, respectively [113]. It is well established that a polarization toward classically activated macrophages aggravates insulin tolerance [114].”(Lines 453-456).

Comment 16 (Energy metabolism and metabolic disorders section): Line 489. The authors could explain what is the consequence of histone H4 acetylation and histone H3 lysine 4 methylation.

Our Response: In this version, we have described the consequences of histone H4 acetylation and histone H3 lysine 4 methylation in transcriptional control as follows:

“Histone H4 acetylation causes conformational changes in the histone structure and facilitates the access of transcriptional factors to gene promoters [120]. Moreover, H3 lysine 4 methylation activates promoters or enhancers through interaction with transcriptional factor complexes, as well as with modifying enzymes for histone H4 acetylation [120,121]. Coinciding with these actions, USP2 knockdown increased histone accessibility, suggesting that USP2 epigenetically impedes the expression of adipose inflammation-associated genes.” (Lines 479-484)

Comment 17 (Nervous system section): Line 540. Explain the reward and punishment systems is and why it is expected to be controlled by USP2.

Our Response: The possible involvement of USP2 in the reward system was suggested in the discussion section of the cited paper [129]. The authors assumed participation of USP2 in the reward system because of the high expression of USP2 in the striatum in mice. In this version of our manuscript, we avoid mentioning the role of USP2 in the reward system entirely, since the assumption was only based on expression data.

Comment 18 (Skeletal and cardiac muscle section): Line 589. Provide example of proinflammatory stimulation that activate USP2 in cardiac tissue.

Our Response: Regulation of USP2 activity by certain proinflammatory stimuli is only speculation based on the fact that USP2 did not exert any visible effects in cardiac tissue, despite its significant expression. As far as we know, there have been no obvious reports identifying the molecules that control USP2 activity under inflammatory conditions. To avoid information overload, we have omitted the sentence including the speculation.

Comment 19 (Immune and inflammatory responses section): Figure 3. Whether cell lines have been assessed (MQ and T cells) this could be indicated in the figure.

Our Response: We now clearly show the information about the experimental models, including the names of cell lines or isolated cells, in Figure 3 of the current version.

Comment 20 (Immune and inflammatory responses section): Line 647. Add RAW264.7 to the sentence: “…stabilizes TNF- α in [RAW264.7] macrophages.”

Our Response: According to the reviewer’s suggestion, we have added “RAW264.7” to the sentence.

“USP2 stabilizes TNF-α in RAW264.7 macrophage-like cells.” (Line 638)

Comment 21 (Male genital tract section): Line 781 and 783. “Usp2KO mouse sperm” could be rephrased as the USP2 deficiency was limited to myeloid cells in the study.

Our Response: As the reviewer pointed out, the aberrant response emerged in sperm isolated from macrophage-selective Usp2KO (msUsp2KO) mice. Therefore, we have replaced “Usp2KO” with “msUsp2KO” in the current version of the manuscript (Lines 765, 772, and 774), and have added this term to the abbreviation list.

Comment 22 (Perspective section): The authors have stated that USP2 have a significant role in tumour-development and this is mentioned in the abstract. The authors could include a sentence, mentioning this.

Our Response: We have added a new paragraph describing USP2-targeting drugs in the Perspectives section of this version. In this paragraph, we mainly focus on anti-tumor drugs that putatively target USP2.

“As described in this review, USP2 has pathophysiological roles in various tissues. Since the roles of USP2 are closely associated with pandemic disorders, including cancer and metabolic diseases, clinical and preclinical studies targeting USP2-associated signaling are desirable. In particular, since USP2 modulates the stability of critical tumor-associated proteins such as cyclin D1, Mdm2, and FASN, great efforts have been made to establish a USP2-targeting anticancer drug. ML364 was reported as a small molecule inhibitor for USP2 and has been proven to evoke cell cycle arrest, cyclin D1 degradation, and inhibition of homologous recombination-mediated DNA repair [42]. Additionally, several studies have reported other chemical inhibitors for USP2, such as isoquinoline-1,3-dione-based compounds [175], chalcone-based compounds [176], 5-(2-thienyl)-3-isoxazoles [43], and a lithocholic acid derivative [40]. Furthermore, other reports have demonstrated that previously established anti-cancer drugs inhibit USP2 activity. The ortho-quinone natural product beta-lapachone, which has been subjected to phase II clinical trials for cancer therapy, was shown to target USP2, resulting in induction of apoptosis in the DU-145 prostate cancer cell line [177]. Moreover, a popular leukemia drug, 6-thioguanine, was shown to be a noncompetitive inhibitor for human USP2 [178]. In addition to evaluation of these drugs for anti-tumor therapy, these drugs could also be tested as treatments for other diseases where the pathology includes contributions from USP2.”(Lines 835-850)

Comment 23 (Perspective section): Whether overexpression of USP2 could result in unspecific off-target deubiquitination (also in mice) is something the authors could consider.

Our Response: In accordance with the reviewer’s comment, we have added a sentence describing the possibility of nonspecific off-target deubiquitination in the cells and animals in the second paragraph of the Perspectives section.

“Moreover, excess USP2 may cause nonspecific off-target deubiquitination in cells and animals.”(Lines 818-819)

Reviewer 2 Report

  1. Authors need to give some details regarding to previous review articles in the similar topic (if any), and compared to current review (highlight the novelty of current review article).
  2. Authors categorized this review into several sections, including Tumorigenesis, Noncancerous proliferation, Apoptosis and autophagy, Circadian clock, Renal system, Energy metabolism and metabolic disorders, Nervous system, Skeletal and cardiac muscles, Immune and inflammatory responses, Male genital tract. By the end (may be in the conclusion section) would be advisable to draw up a figure to illustrate USP2-related signaling pathways more systematically. Readers would be easier to appreciate the biological function roles of USP2-related mechanisms in vivo!
  3. Would be helpful to have a "brief" summary section to clearly illustrate and summarize the whole (or part of) commentary.
  4. Need to discuss if any drugs (used clinically or currently under clinical trails) which are targeting USP2-associated signaling. 

Author Response

Reviewer#2

I appreciate the reviewer’s comments, and have revised the manuscript accordingly. My responses to the comments are provided below.

Comment 1: Authors need to give some details regarding to previous review articles in the similar topic (if any), and compared to current review (highlight the novelty of current review article).

Our Response: Previous reviews have focused on more specific topics, such as cancer-associated signaling, muscle atrophy, sodium channel regulation, and expressional control of USP2 variants. Compared with these examples, our review deals with cell signaling and its outcome in a more comprehensive way. In the current manuscript version, we now indicate this as follows:

“By comparing USP2 with other DUBs, some review articles have summarized the pathological roles of USP2 in specific functional areas, such as cancer promotion [14,15], muscle atrophy modification [16], and sodium channel regulation [17]. Additionally, another review focused on the expressional control of alternative splicing variants of USP2 [8]. Despite these examples in the literature, there have been no review articles summarizing USP2-associated signaling and its outcomes from a more comprehensive viewpoint. In this review, therefore, we examine the pathophysiological events elicited by USP2 and summarize the cellular signaling that underlies the events in a comprehensive way.” (Lines 56-63).

Comment 2: Authors categorized this review into several sections, including Tumorigenesis, Noncancerous proliferation, Apoptosis and autophagy, Circadian clock, Renal system, Energy metabolism and metabolic disorders, Nervous system, Skeletal and cardiac muscles, Immune and inflammatory responses, Male genital tract. By the end (may be in the conclusion section) would be advisable to draw up a figure to illustrate USP2-related signaling pathways more systematically. Readers would be easier to appreciate the biological function roles of USP2-related mechanisms in vivo!

Our Response: We completely agree with the reviewer’s comment since this review covers a wide variety of USP2 functions in vivo. We have now provided a summarization figure as a graphical abstract. Additionally, we have added Table 1, which summarizes the functions of USP2 in vivo and in vitro, in the Conclusion section.

Comment 3: Would be helpful to have a "brief" summary section to clearly illustrate and summarize the whole (or part of) commentary.

Our Response: In accordance with the reviewer’s comment, we have added a brief summary and conclusion section at the end of the manuscript.

“The findings detailed in this article are summarized in Table 1. USP2, mainly USP2-1, participates in tumorigenesis in various cancers through the potentiation of the cell cycle, mitosis, lipogenesis, metastasis, EMT, and anti-oxidation, while inhibiting p53-mediated tumor death. USP2 also controls TNF-elicited apoptosis as well as TNF resistance. Additionally, cFLIP-regulated autophagy is also regulated by USP2. USP2, especially USP2-4, is a core component of the biological clock, and contributes to the circadian rhythm in the liver and SCN by deubiquitinating BMAL1 and PER1. In the intestine, USP2-4 determines rhythmic calcium absorption, resulting in the maintenance of calcium homeostasis. Additionally, in vitro data demonstrates crucial roles of USP2-4 in ENaC and MR expression, suggesting USP2 controls sodium adsorption, although Usp2KO mice presented no obvious phenotypic changes, including blood pressure. Accumulating evidence indicates that USP2 is a determinant for energy metabolism in hepatocytes, myoblasts, and cancerous cells. Moreover, previous reports also suggest significant roles of USP2 in the intercommunications between energy-competent organs such as liver, skeletal muscle and adipose tissue. Thus, aberrant expression of USP2 is considered to provoke metabolic diseases, such as type 2 diabetes and atherosclerosis. USP2 also maintains neural activity in the brain, and is postulated to be involved in special memory retrieval, motor coordination, short-term recognition, sensorimotor gating, and anxiety-like behavior. In terms of skeletal and cardiac muscles, USP2 is considered to modify differentiation, contraction, and remodeling after pressure overloads, although Usp2KO mice displayed no obvious phenotypic changes under normal conditions. At the present time, the function of USP2 in immunoregulation is controversial. Some reports have demonstrated that USP2 stimulates the production of cytokines and anti-viral proteins through NF-B and STAT1-dependent mechanisms. In contrast, USP2 has also been suggested to attenuate the production of cytokines and microbicide peptides by perturbing the OCT1/2, NF-B, and IRF3 pathways. In agreement with the observation that USP2 is most abundantly expressed in testis, USP2 deficiency caused severe male sterility due to defects in sperm motility. This malfunction of sperm is accounted for by the lack of USP2 in sperm and male genital tract macrophages. Although previous efforts have clarified a wide variety of roles played by USP2, the detailed molecular mechanisms underlying these phenomena remain to be uncovered in further studies. Given that there are several discrepancies between experimental models, reevaluation of data by modern sophisticated and comprehensive techniques is also required.” (Lines 871-899)

Comment 4: Need to discuss if any drugs (used clinically or currently under clinical trails) which are targeting USP2-associated signaling. 

Our Response: We have added a new paragraph describing USP2-targeting drugs to the Perspectives section of this version. In this paragraph, we introduce some USP2 inhibitors obtained by chemical screening to generate novel anti-cancer drugs. As far as we know, these drugs have not yet been applied as clinical or preclinical treatments. We also refer to two drugs, beta-lapachone and 6-thioguanine, both of which are widely known to have anti-tumor activity. In this version we mention these drugs potentially target USP2. 

“As described in this review, USP2 has pathophysiological roles in various tissues. Since the roles of USP2 are closely associated with pandemic disorders, including cancer and metabolic diseases, clinical and preclinical studies targeting USP2-associated signaling are desirable. In particular, since USP2 modulates the stability of critical tumor-associated proteins such as cyclin D1, Mdm2, and FASN, great efforts have been made to establish a USP2-targeting anticancer drug. ML364 was reported as a small molecule inhibitor for USP2 and has been proven to evoke cell cycle arrest, cyclin D1 degradation, and inhibition of homologous recombination-mediated DNA repair [42]. Additionally, several studies have reported other chemical inhibitors for USP2, such as isoquinoline-1,3-dione-based compounds [175], chalcone-based compounds [176], 5-(2-thienyl)-3-isoxazoles [43], and a lithocholic acid derivative [40]. Furthermore, other reports have demonstrated that previously established anti-cancer drugs inhibit USP2 activity. The ortho-quinone natural product beta-lapachone, which has been subjected to phase II clinical trials for cancer therapy, was shown to target USP2, resulting in induction of apoptosis in the DU-145 prostate cancer cell line [177]. Moreover, a popular leukemia drug, 6-thioguanine, was shown to be a noncompetitive inhibitor for human USP2 [178]. In addition to evaluation of these drugs for anti-tumor therapy, these drugs could also be tested as treatments for other diseases where the pathology includes contributions from USP2.”(Lines 835-850)